# Structural basis of tubulin recruitment and assembly by microtubule polymerases with tumor overexpressed gene (TOG) domain arrays

Stanley Nithianantham[1], Brian D Cook[1], Madeleine Beans[1], Fei Guo[1], Fred Chang[2], Jawdat Al-Bassam[1]*

[1]Molecular Cellular Biology Department, University of California, Davis, United States; [2]Department of Cell and Tissue Biology, University of California, San Francisco, United States

**Abstract** XMAP215/Stu2/Alp14 proteins accelerate microtubule plus-end polymerization by recruiting tubulins via arrays of tumor overexpressed gene (TOG) domains, yet their mechanism remains unknown. Here, we describe the biochemical and structural basis for TOG arrays in recruiting and polymerizing tubulins. Alp14 binds four tubulins via dimeric TOG1-TOG2 subunits, in which each domain exhibits a distinct exchange rate for tubulin. X-ray structures revealed square-shaped assemblies composed of pseudo-dimeric TOG1-TOG2 subunits assembled head-to-tail, positioning four unpolymerized tubulins in a polarized wheel-like configuration. Crosslinking and electron microscopy show Alp14-tubulin forms square assemblies in solution, and inactivating their interfaces destabilize this organization without influencing tubulin binding. An X-ray structure determined using approach to modulate tubulin polymerization revealed an unfurled assembly, in which TOG1-TOG2 uniquely bind to two polymerized tubulins. Our findings suggest a new microtubule polymerase model in which TOG arrays recruit tubulins by forming square assemblies that then unfurl, facilitating their concerted polymerization into protofilaments.
DOI: https://doi.org/10.7554/eLife.38922.001

*For correspondence:
jawdat@ucdavis.edu

Competing interests: The authors declare that no competing interests exist.

## Introduction

Microtubules (MTs) are highly dynamic polarized polymers that perform critical and diverse cellular functions including formation of bipolar mitotic spindles, intracellular organization, and modulation of cell development and cell migration (*Akhmanova and Steinmetz, 2008*; *Akhmanova and Steinmetz, 2015*). MTs are assembled from αβ-tubulin heterodimers (herein termed αβ-tubulin), and their polymerization exhibits dynamic instability arising from guanosine triphosphate (GTP) hydrolysis in β-tubulins at MT ends. However, the conformational changes promoting soluble αβ-tubulins to polymerize at MT ends remain poorly understood. Polymerization of αβ-tubulin and GTP hydrolysis are regulated by conserved proteins that bind at MT plus-ends or along MT lattices (*Akhmanova and Steinmetz, 2008*; *Akhmanova and Steinmetz, 2011*; *Akhmanova and Steinmetz, 2015*; *Al-Bassam and Chang, 2011*; *Al-Bassam et al., 2010*; *Brouhard and Rice, 2014*). The XMAP215/Stu2/Alp14 MT polymerases are among the best-studied families of MT regulators. They localize to the extreme tips of MT plus-ends and accelerate αβ-tubulin polymerization in eukaryotes (*Akhmanova and Steinmetz, 2011*; *Akhmanova and Steinmetz, 2015*; *Al-Bassam and Chang, 2011*; *Maurer et al., 2014*). Loss or depletion of MT polymerases is lethal in most eukaryotes as it severely decreases MT polymerization rates during interphase, resulting in shortened mitotic spindles (*Al-Bassam et al., 2012*; *Cullen et al., 1999*; *Wang and Huffaker, 1997*). MT polymerases also

bind kinetochores, where they accelerate MT dynamics and regulate kinetochore-MT attachment (*Miller et al., 2016*; *Tanaka et al., 2005*). MT polymerases recruit αβ-tubulins via arrays of conserved tumor overexpressed gene (TOG) domains (herein termed TOG arrays), which are critical for their function (*Reber et al., 2013*; *Widlund et al., 2011*). Arrays of TOG-like domains are conserved in two other classes of MT regulators, CLASP and Crescerin/CHE-12 protein families (*Al-Bassam and Chang, 2011*; *Al-Bassam et al., 2010*; *Das et al., 2015*), suggesting that arrays of TOG domains uniquely evolved to regulate diverse MT polymerization functions through the binding of αβ-tubulins in various intracellular settings.

Yeast MT polymerases, such as *Saccharomyces cerevisiae* Stu2p and *Schizosccharomyces pombe* Alp14, are homodimers containing two unique and conserved TOG domain classes, TOG1 and TOG2, per subunit, numbered based on their proximity to the N-terminus in the protein sequence. In contrast, metazoan orthologs, such as XMAP215 and ch-TOG, are monomers with five tandem TOG domains, TOG1 through TOG5 (*Al-Bassam and Chang, 2011*; *Brouhard and Rice, 2014*). Phylogenetic analyses suggest that TOG1 and TOG2 domains are evolutionarily distinct (*Al-Bassam and Chang, 2011*), and that TOG3 and TOG4 domains in metazoans are evolutionarily and structurally exclusively related to the TOG1 and TOG2 domains, respectively (*Brouhard et al., 2008*; *Fox et al., 2014*; *Howard et al., 2015*). Thus, despite differences in TOG array organization in yeast and metazoan proteins, both groups contain an array of tandem TOG1-TOG2 domains.

Structural studies contributed to our understanding of the molecular basis of TOG domain function in recruiting soluble αβ-tubulin. Each TOG domain is composed of six α-helical HEAT (Huntingtin, EF3A, ATM, and TOR) repeats, which form a conserved paddle-shaped structure (*Al-Bassam and Chang, 2011*; *Al-Bassam et al., 2007*; *Al-Bassam et al., 2006*; *Brouhard and Rice, 2014*; *Slep and Vale, 2007*). X-ray structures of isolated TOG1 and TOG2 domains in complex with αβ-tubulins revealed that these domains recognize the curved αβ-tubulin conformations via inter-helical loops positioned along an edge of these paddle-shaped domains (*Ayaz et al., 2014*; *Ayaz et al., 2012*). Straightening of the curved αβ-tubulins upon polymerization into MTs likely dissociates TOG domains from the complexes. Our previous studies indicate that native TOG arrays from yeast or metazoan MT polymerases assemble into discrete particle-like assemblies upon binding αβ-tubulin (*Al-Bassam and Chang, 2011*; *Al-Bassam et al., 2006*). Both TOG1 and TOG2 domains are critical for MT polymerase function, and their inactivation results in MT functional defects (*Al-Bassam et al., 2012*; *Al-Bassam et al., 2006*; *Ayaz et al., 2014*). Two models were proposed to explain how arrays of TOG domains function as MT polymerases. One model based on studies of native TOG arrays indicates that they may form ordered assemblies upon binding αβ-tubulins (*Al-Bassam et al., 2006*; *Brouhard et al., 2008*). A second model, based on studies of isolated TOG domains or short TOG arrays, suggests that these arrays form flexible assemblies in which TOG1 and TOG2 independently recruit multiple αβ-tubulins to MT plus-ends (*Al-Bassam and Chang, 2011*; *Ayaz et al., 2014*). Distinguishing between these models requires understanding the high-resolution organization of native TOG arrays in complex with αβ-tubulin and their transitions during αβ-tubulin recruitment and polymerization phases.

Here, we describe biochemical and structural states of TOG arrays during αβ-tubulin recruitment and polymerization states. We show that the yeast MT polymerase, Alp14, recruits four αβ-tubulins using dimeric arrays of TOG1-TOG2 domains. TOG1 and TOG2 domains each bind and release αβ-tubulins with different rates. X-ray structures reveal pseudo-dimeric TOG1-TOG2 subunits in head-to-tail square-shaped assemblies, each of which orients four unpolymerized αβ-tubulins in a polarized configuration. Crosslinking and mass spectrometry and electron microscopy studies show that dimeric yeast TOG arrays form these square assemblies in solution. Alp14 mutants with inactivated binding interfaces show disorganized configurations or polymerized arrangements, but without any defects in αβ-tubulin binding. Using a novel approach to promote the limited polymerization of αβ-tubulins while bound to TOG arrays, we determined an X-ray structure of an 'unfurled' TOG1-TOG2 array:αβ-tubulin assembly revealing TOG1 and TOG2 bound to two αβ-tubulins polymerized head-to-tail into a protofilament. Our studies establish a new 'polarized unfurling' model for TOG arrays as MT polymerases.

## Results

### TOG1 and TOG2 domains possess distinct affinities for αβ-tubulin

The *S. pombe* Alp14 is a typical yeast MT polymerase, consisting of a tandem array of N-terminal TOG1 and TOG2 domains separated by a 60-residue linker, followed by a Ser-Lys-rich (SK-rich) region and a C-terminal coiled-coil domain that regulates dimerization (*Figure 1A*). We studied the αβ-tubulin binding capacities and stoichiometries of near native monomeric and dimeric Alp14, which both consist of TOG1-TOG2 arrays and differ by the presence of a C-terminal SK-rich region and dimerization coiled-coil domain. Using quantitative size-exclusion chromatography (SEC) with multi-angle light scattering (SEC-MALS), we measured the αβ-tubulin binding stoichiometry for monomeric Alp14 (herein termed Alp14-monomer: residues 1–510), and dimeric Alp14 (herein termed wt-Alp14-dimer: residues 1–690) at 80–100 mM KCl ionic strength (*Figure 1A–B*, *Figure 1D*; details described in *Figure 1—figure supplement 1A–C,D–F*; SEC-MALS control experiments shown in *Figure 1—figure supplement 2G–H*; *Table 1*) (*Al-Bassam et al., 2012*). We show that 1 μM wt-Alp14-dimer binds four αβ-tubulins per dimer via its four TOG domains (dimeric TOG1-TOG2 arrays), whereas 1 μM wt-Alp14-monomer binds two αβ-tubulins per dimer via two TOG domains. Thus, TOG1 and TOG2 independently recruit αβ-tubulins (*Figure 1A–B,D*; *Figure 1—figure supplement 1A–C,D–F*; *Tables 1* and *2*).

Since TOG1 and TOG2 domains bind αβ-tubulins via narrow and mostly ionic binding interfaces, we studied the effect of a moderate increase in ionic strength (100–200 mM KCl) on αβ-tubulin binding capacities of TOG1 and TOG2 domains in these arrays (*Ayaz et al., 2014*; *Ayaz et al., 2012*). At 200 mM KCl, both 1 μM wt-Alp14-monomer and wt-Alp14-dimer bound roughly half as much αβ-tubulin than at 80–100 mM KCl. Thus, either TOG1 or TOG2 domains may lose part or all of their αβ-tubulin-binding capacity at 200 mM KCl (*Figure 1B,D*; *Figure 1—figure supplement 1A–C,D–F* and *Tables 1* and *2*). These differences between the αβ-tubulin-binding stoichiometries at 100 versus 200 mM KCl resolve discrepancies regarding Alp14- or Stu2-αβ-tubulin-binding stoichiometries reported previously (*Ayaz et al., 2014*; *Al-Bassam et al., 2012*; *Al-Bassam et al., 2006*).

We next determined if the αβ-tubulin-binding capacity of TOG1 or TOG2 changed within arrays due to a change in ionic strength from 100 to 200 mM KCl. Using SEC and SEC-MALs, we studied the αβ-tubulin-binding stoichiometry for Alp14-dimer mutants in which either TOG1 (termed TOG1M) or TOG2 (termed TOG2M) were inactivated through multiple-residue mutations in the αβ-tubulin-binding interfaces (*Al-Bassam et al., 2007*; *Ayaz et al., 2012*); see Materials and methods; *Figure 1C*, *Figure 1D*; *Figure 1—figure supplement 1G–L*). 1 μM TOG1M, which only includes two active TOG2 domains, bound two αβ-tubulins at 100 mM KCl, but almost completely dissociated from αβ-tubulin at 200 mM KCl. TOG1M and αβ-tubulin did not co-migrate on SEC, and most of the αβ-tubulin migrated as a separate peak at 200 mM KCl (*Figure 1D*, *Figure 1—figure supplement 1G–I*; *Tables 1* and *2*). In contrast, TOG2M, which only includes two active TOG1 domains, bound and co-migrated with αβ-tubulin in both 100 and 200 mM KCl conditions (*Figure 1D*, *Figure 1—figure supplement 1J–L*, *Tables 1* and *2*). Moreover, molar ratios of αβ-tubulin bound to TOG1M and TOG2M measured by quantitative-SEC at 100 mM KCl and maximal αβ-tubulin stoichiometry determined by SEC-MALS at 80 mM KCl were roughly half of that measured for wt-Alp14-dimer at 80–100 mM KCl (*Figure 1C,D*). These data support the independent and distinct affinities of TOG1 and TOG2 domains in recruiting αβ-tubulins while within either monomeric or dimeric arrays, and that dimerization does not change the αβ-tubulin-binding stoichiometry. TOG1 and TOG2 showed distinct αβ-tubulin binding behaviors at 100 and 200 mM KCl (*Figure 1F*; the SEC-MALS controls are shown in *Figure 1—figure supplement 2G*)

Next, we quantitatively measured the absolute TOG1 and TOG2 binding affinities for αβ-tubulin using isothermal titration calorimetery (ITC), and determined how these affinities were influenced by changes in ionic strength at 100–200 mM KCl. ITC data showed that isolated TOG1 (residues 1–270) and TOG2 (residues 320–510) bound αβ-tubulins with roughly 2.5-fold difference in dissociation constants (*Figure 1E*; *Figure 1—figure supplement 3*). At 100 mM KCl, the dissociation constants for TOG1 and TOG2 were measured at 70 and 173 nM, respectively. These data suggest that both TOG1 and TOG2 exhibit a moderately high affinity for αβ-tubulin with a 2.5-fold difference in affinity, nearly identical to that previously reported (*Ayaz et al., 2014*; *Ayaz et al., 2012*). However, at 200 mM KCl, we measured TOG1 and TOG2 αβ-tubulin dissociation constants at 1.5 μM and 3.2 μM, respectively, which showed 20-fold weaker affinity in absolute values compared to those

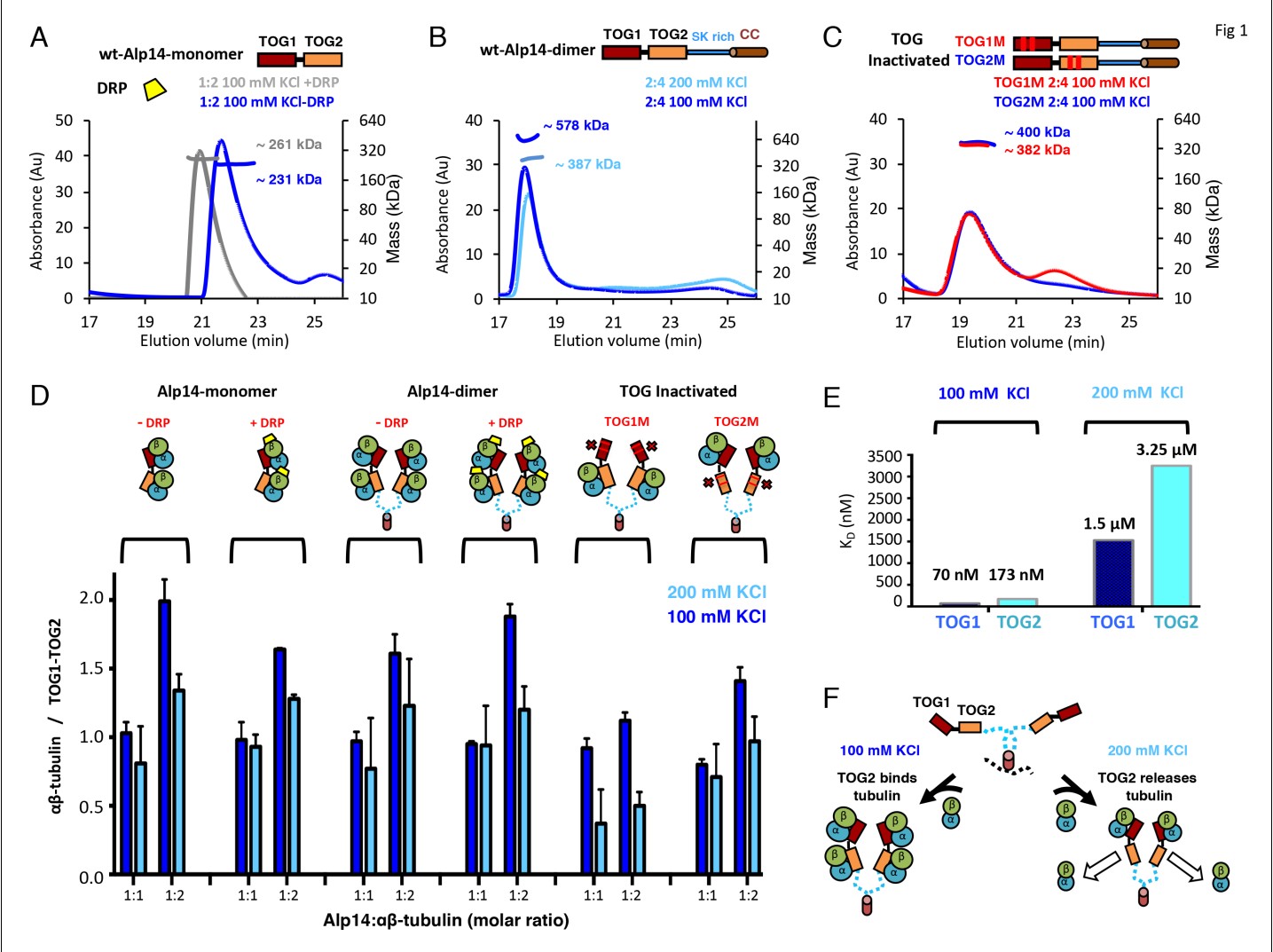

**Figure 1.** TOG1 and TOG2 domains bind αβ-tubulins and exchange them at different rates, within Alp14 TOG arrays. (**A**) Top, domain organization of Alp14-monomer. Bottom, SEC-MALS for wt-Alp14-monomer at 100 mM KCl with and without DRP, revealing two αβ-tubulins bound in a non-polymerized state. (**B**) Top, domain organization of Alp14-dimer. Bottom, SEC-MALS for wt-Alp14-dimer in αβ-tubulin complex at 2:4 stoichiometry at 100 mM and 200 mM KCl, revealing that four αβ-tubulins bind dimeric-Alp14 with four TOG domains and two αβ-tubulins dissociate upon increase of ionic strength (masses reported in *Table 2*). (**C**) Top, organization of TOG-inactivated Alp14-dimer (C: TOG1M, TOG2M). Bottom, SEC-MALS of 4:2 TOG1M- and TOG2M-αβ-tubulin at 100 mM KCl revealing half of the αβ-tubulin binding stoichiometry compared to wt-Alp14-dimer (shown in B). (**D**) SEC-based titration measured through analyses of masses from SDS-PAGE (*Figure 1—figure supplement 1*; Materials and methods) of Alp14-monomer, Alp14-dimer, and TOG1M and TOG2M with and without DRP binding, reveals the binding capacities of TOG1 and TOG2 domains in these constructs and their non-equivalence in αβ-tubulin exchange at 200 mM KCl. DRP binding to αβ-tubulin does not influence αβ-tubulin binding to TOG arrays. Details are described in *Figure 1—figure supplement 1–2*. (**E**) Isothermal titration calorimetery (ITC) reveals TOG1 and TOG2 exchange αβ-tubulin with non-equivalent rates at 200 mM KCl. At 100 mM KCl, TOG1 and TOG2 both slowly dissociate from αβ-tubulin with $K_D$ = 70 nM and $K_D$ = 173 nM, respectively. At 200 mM KCl, TOG1 dissociates from αβ-tubulin slowly with a $K_D$ = 1.50 μM while TOG2 mostly dissociates from αβ-tubulin with a $K_D$ = 3.25 μM. ITC binding curves are shown in *Figure 1—figure supplement 3*. (**F**) Model for non-equivalent activities of TOG1 and TOG2 within TOG array for recruiting αβ-tubulins.

DOI: https://doi.org/10.7554/eLife.38922.002

The following figure supplements are available for figure 1:

**Figure supplement 1.** The Alp14 TOG array:αβ-tubulin binding capacities reveal non-equivalent behavior of TOG1 and TOG2 in different conditions.
DOI: https://doi.org/10.7554/eLife.38922.003

**Figure supplement 2.** Molar ratios of Alp14 constructs binding to αβ-tubulin binding in the presence of Darpin-D1 (DRP) show no change in stoichiometry, and control SEC-MALS traces.
DOI: https://doi.org/10.7554/eLife.38922.004

*Figure 1 continued on next page*

*Figure 1 continued*

**Figure supplement 3.** Isothermal titration calorimetry (ITC) reveals Alp14-TOG1 and TOG2 αβ-tubulin-binding affinities.
DOI: https://doi.org/10.7554/eLife.38922.005

measured at 100 mM KCl. Together, our studies suggest that within the cellular αβ-tubulin concentration range (5–10 µM) and 1 µM Alp14 at 100 mM KCl or below, each TOG1-TOG2 subunit tightly binds two αβ-tubulins, whereas at 200 mM KCl, each TOG1-TOG2 subunit binds one αβ-tubulin tightly via TOG1 and exchanges a second αβ-tubulin rapidly via TOG2. In conditions of 200 mM KCl, 1 µM Alp14, and 5–10 µM αβ-tubulin, TOG1 domains is almost completely occupied by αβ-tubulin, while TOG2 domains is mostly dissociated from αβ-tubulin (*Figure 1F*).

## X-ray structures of a recruitment complex: pseudo-dimeric TOG1-TOG2 arrays in a head-to-tail square assembly that pre-orients αβ-tubulins

Our biochemical analyses suggest that structural studies of TOG array:αβ-tubulin complexes must be conducted at lower ionic strengths of 80–100 mM KCl and at high αβ-tubulin concentrations to avoid αβ-tubulin dissociation from TOG2 domains. To further increase TOG1-TOG2:αβ-tubulin complex formation and inhibit αβ-tubulin self-assembly under such conditions, we utilized the designed ankyrin repeat protein (DARPin) D1 (herein termed DRP), which specifically binds and neutralizes the β-tubulin polymerizing interface (*Pecqueur et al., 2012*). First, we studied if DRP binding to αβ-tubulin influenced wt-Alp14-monomer or wt-Alp14-dimer binding stoichiometries to multiple αβ-tubulins. We measured binding molar ratios using quantitative-SEC and stoichiometry using SEC-MALs for DRP-bound-αβ-tubulin to wt-Alp14-dimer and wt-Alp14-monomer in 80–100 mM KCl conditions, respectively (*Figure 1A*; *Figure 1—figure supplement 2A–F*; *Tables 1* and *2*). We showed that DRP did not affect the simultaneous binding of multiple αβ-tubulins to TOG1-TOG2 arrays in either wt-Alp14-monomer or wt-Alp14-dimer. 1 µM wt-Alp14-dimer formed a complex with four αβ-tubulins and four DRPs at a molar ratio of 2:4:4 (*Figure 1D*; *Figure 1—figure supplement 2D–F*). We measured a mass of the wt-Alp14-monomer:αβ-tubulin:DRP complex by SEC-MALS that indicated a 1:2:2 stoichiometry complex in which each TOG1-TOG2 subunit bound two αβ-tubulins, each of which bound its own DRP (*Figure 1A,D*; *Figure 1—figure supplement 2A–C,G*; *Table 2*). The ability

**Table 1.** The stoichiometry for MT polymerases TOG1-TOG2 binding αβ-tubulin and DARPin (DRP)

| Protein complex | Expected Mass (kDa) | SEC-MALS Measured Mass (kDa) | SEC elution volume (mL) | Apparent Mass (kDa) |
|---|---|---|---|---|
| αβ-Tubulin | 100 kDa | 98 ± 0.323* | 12.9 | 104 |
| Alp14-dimer | 150 kDa | 143 ± 1.70 | 10.0 | 675 |
| 1 Alp14-dimer: 1 Tubulin 80 mM KCl | 350 kDa | 387 ± 2.74 | 9.27 | 463 |
| 1 Alp14-dimer: 2 Tubulin 80 mM KCl | 550 kDa | 578 ± 1.41 | 8.98 | 784 |
| 1 Alp14-dimer: 1Tubulin 200 mM KCl | 350 kDa | 304 ± 11.8 | 9.22 | 693 |
| 1 Alp14-dimer: 2 Tubulin 200 mM KCl | 350 kDa | 392 ± 9.66 | 9.67 | 549 |
| 1 Alp14-dimer-TOG1M: 1 Tubulin 80 mM KCl | 350 kDa | 400 ± 7.3 | 10.13 | 433 |
| 1 Alp14-dimer-TOG2M: 2 Tubulin 80 mM KCl | 350 kDa | 382 ± 14 | 9.76 | 524 |
| Alp14-monomer | 62 kDa | 77.8 ± 1.21 | 12.98 | 99 |
| 1 Alp14-monomer: 2 Tubulin 80 mM KCl | 262 kDa | 264 ± 1.31 | 11.17 | 253 |
| 1 Alp14-monomer: 2 Tubulin: 2 DRP 80mM KCl | 298 kDa | 312 ± 2.32 | 10.94 | 285 |
| 1 Alp14-dimer-INT1: 2 Tubulin 80mM KCl | 550 kDa | 533 ± 3.11 | 8.95 | 775 |
| 1 Alp14-dimer-INT2: 2 Tubulin 80mM KCl | 550 kDa | 580 ± 3.11 | 8.90 | 770 |
| 1 Alp14-dimer-INT1+2: 2 Tubulin 80mM KCl | 550 kDa | 540 ± 3.22 | 8.80 | 790 |

*Standard error is defined based on fitting data across peaks using Astra-software.

DOI: https://doi.org/10.7554/eLife.38922.006

**Table 2.** Capacities of MT polymerase TOG1-TOG2 to bind αβ-tubulin, influenced by ionic strength

| Protein Complex (Alp14-monomer concentration 4.43 uM) | μM tubulin* bound | μM tubulin free | TOG1-TOG2 :αβ-tubulin |
|---|---|---|---|
| wt-Alp14-monomer (2 μM) | | | |
| 1 Alp14-monomer: 1 Tub 100 mM KCl | 4.57 ± 0.08 | 0.43 ± 0.08 | 1.04 |
| 1 Alp14-monomer: 2 Tub 100 mM KCl | 8.82 ± 0.16 | 1.18 ± 0.16 | 2 |
| 1 Alp14-monomer: 1 Tub 200 mM KCl | 3.37 ± 0.27 | 1.63 ± 0.27 | 0.76 |
| 1 Alp14-monomer: 2 Tub 200 mM KCl | 5.97 ± 0.12 | 4.03 ± 0.12 | 1.35 |
| 1 Alp14-monomer:1 Tub:1 DRP 100 mM KCl | 4.33 ± 0.13 | 0.67 ± 0.13 | 0.98 |
| 1 Alp14-monomer:2 Tub:2 DRP 100 mM KCl | 7.23 ± 0.01 | 2.77 ± 0.01 | 1.64 |
| 1 Alp14-monomer:1 Tub:1 DRP 200 mM KCl | 4.13 ± 0.09 | 0.87 ± 0.09 | 0.94 |
| 1 Alp14-monomer: 2 Tub 2 DRP 200 mM KCl | 5.67 ± 0.03 | 4.34 ± 0.03 | 1.28 |
| wt-Alp14-dimer (1 μM) | | | |
| 1 Alp14-dimer: 1 Tub 100 mM KCl | 4.31 ± 0.07 | 0.69 ± 0.07 | 0.98 |
| 1 Alp14-dimer: 2 Tub 100 mM KCl | 7.11 ± 0.14 | 2.89 ± 0.14 | 1.61 |
| 1 Alp14-dimer: 1 Tub 200 mM KCl | 3.44 ± 0.37 | 1.56 ± 0.37 | 0.78 |
| 1 Alp14-dimer: 2 Tub 200 mM KCl | 5.46 ± 0.34 | 4.55 ± 0.34 | 1.24 |
| 1 Alp14-dimer: 1 Tub: 2 DRP 100 mM KCl | 4.23 ± 0.02 | 0.78 ± 0.02 | 0.96 |
| 1 Alp14-dimer: 1 Tub: 2 DRP 100 mM KCl | 8.33 ± 0.09 | 1.67 ± 0.09 | 1.89 |
| 1 Alp14-dimer: 1 Tub: 1 DRP 200 mM KCl | 4.17 ± 0.29 | 0.83 ± 0.29 | 0.95 |
| 1 Alp14-dimer: 2 Tub: 2 DRP 200 mM KCl | 6.02 ± 0.17 | 3.99 ± 0.17 | 1.36 |
| TOG square Interface mutants (1 μM) | | | |
| 1 INT1: 2 Tub 100mM KCl | 8.34 ± 0.07 | 1.66 ± 0.07 | 1.74 |
| 1 INT1: 2 Tub 200mM KCl | 5.67 ± 0.04 | 4.33 ± 0.04 | 1.18 |
| 1 INT2: 2 Tub 100mM KCl | 9.52 ± 0.3 | 0.48 ± 0.2 | 2.02 |
| 1 INT2: 2 Tub 200mM KCl | 6.82 ± 0.2 | 3.1 ±0.1 | 1.45 |
| 1 INT1+2: 2 Tub 100mM KCl | 8.44 ± 0.05 | 1.56 ± 0.05 | 1.86 |
| 1 INT1+2: 2 Tub 200mM KCl | 6.85 ± 0.04 | 3.15 ± 0.04 | 1.46 |
| Inactivated TOG mutants (1 μM) | | | |
| 1 TOG1M: 1 Tub 100 mM KCl | 4.04 ± 0.07 | 0.97 ± 0.07 | 0.92 |
| 1 TOG1M: 2 Tub 100 mM KCl | 4.94 ± 0.06 | 5.06 ± 0.06 | 1.12 |
| 1 TOG1M: 1 Tub 200 mM KCl | 1.66 ± 0.25 | 3.34 ± 0.25 | 0.38 |
| 1 TOG1M: 2 Tub 200 mM KCl | 2.23 ± 0.1 | 7.77 ± 0.1 | 0.51 |
| 1 TOG2M: 1 Tub 100 mM KCl | 3.53 ± 0.04 | 1.48 ± 0.04 | 0.8 |
| 1 TOG2M: 2 Tub 100 mM KCl | 6.24 ± 0.10 | 3.76 ± 0.1 | 1.41 |
| 1 TOG2M: 1 Tub 200 mM KCl | 3.15 ± 0.24 | 1.85 ± 0.24 | 0.72 |
| 1 TOG2M: 2 Tub 200 mM KCl | 4.28 ± 0.18 | 5.72 ± 0.18 | 0.97 |

*Standard error is defined based on combined data from duplicated SEC runs.

DOI: https://doi.org/10.7554/eLife.38922.010

of Alp14-bound αβ-tubulin to bind stoichiometric amounts of DRP suggests that αβ-tubulins recruited by TOG arrays are in a non-polymerized state upon initial association with Alp14. This feature is consistent with a reported lack of cooperativity described between TOG1 and TOG2 in binding to αβ-tubulins (*Ayaz et al., 2014*). Thus, we used this strategy to identify crystallization conditions using TOG array orthologs from a variety of organisms (see Materials and methods).

Crystals of the *Saccharomyces kluyveri* ortholog of Alp14 (herein termed sk-Alp14-monomer: residues 1–550) bound to αβ-tubulins and DRP grew in conditions similar to those used for SEC and SEC-MALS (*Figure 2—figure supplement 1A*). Using crystals with either a native sk-Alp14-monomer

or an sk-Alp14-monomer with a modified TOG1-TOG2 linker sequence (termed sk-Alp14-monomer-SL; see Materials and methods; *Figure 2—figure supplement 2*), we determined X-ray structures for 1:2:2 TOG1-TOG2 array:αβ-tubulin:DRP from complexes using sk-Alp14-monomer and sk-Alp14-monomer-SL using molecular replacement (see Materials and methods) at 4.4 Å and 3.6 Å resolution, respectively (*Table 3* and *Figure 2—figure supplement 1B,C*). In the structures, TOG1 domains were clearly differentiated from TOG2 domains by their conserved C-terminal extension and jutting α-helix that were unambiguously identified in density-modified maps (*Figure 2—figure supplement 1D,E*). Each asymmetric unit contained two wheel-shaped assemblies (*Figure 2—figure supplement 1F*) representing two sets of alternating TOG1 and TOG2 domains oriented in a square-like conformation (termed the TOG square), with each TOG domain binding a DRP-capped αβ-tubulin on its outer edge. Excluding the 10-residue TOG1-TOG2 linker region immediately

**Table 3.** X-ray Crystallographic and Refinement statistics of MT polymerase:αβ-tubulin:DRP.

| Data collection | 1:2:2 sk-Alp14-monomer: αβ-Tubulin:DRP | 1:2:2 sk-Alp14-monomer-SL: αβ-Tubulin:DRP | 1:2:2 sk-Alp14-monomer: αβ-Tubulin:DRPΔN | 1:2:2 sk-Alp14-dimer: αβ-Tubulin:DRPΔN |
|---|---|---|---|---|
| Resolution range (Å) | 96.59 - 4.40 (4.64- 4.40)* | 59.45 – 3.60 (3.79 – 3.60)* | 57.56 – 3.20 (3.37 – 3.20)* | 99.83 – 3.5 (3.69 – 3.50)* |
| Space group | $P\,2_1$ | $P\,2_1$ | $P\,2_1$ | $P\,2_1$ |
| Wavelength (Å) | 0.9792 | 0.9792 | 0.9792 | 0.9792 |
| Unit cell (Å): *a, b, c* (°): β | 218.80, 107.65, 282.74 90.38 | 218.48, 106.15, 282.23 90.39 | 115.13, 194.99, 149.57 90.19 | 122.73 199.67, 162.70 90.09 |
| Total number of observed reflections | 229567 | 380856 | 298551 | 235576 |
| Unique reflections | 80099 {68039}[†] | 142673 {121943}[†] | 104265 {88337}[†] | 91368 |
| Average mosaicity | 0.57 | 0.38 | 0.64 | 0.50 |
| Multiplicity | 2.9 (2.9)* | 2.7 (2.7)* | 2.9 (2.9)* | 2.6 (2.4)* |
| Completeness (%) | 95.4 (94.8) {80.6}[†] | 95.0 (96.7) {79.0}[†] | 96.2 (97.9) {82.0}[†] | 92.9 (90.2)* |
| Wilson B-factor (Å$^2$) | 82.7 | 81.4 | 46.6 | - |
| $<I/\sigma\,(I)>$ | 4.9 (1.9)* | 4.8 (1.2)* | 5.8 (1.5)* | 4.8 (1.1)* |
| $R_{merge}{}^c$ | 0.14 (0.48)* | 0.13(0.65)* | 0.13(0.65)* | 0.14 (0.63)* |
| **Structure refinement** | | | | |
| $R_{work}$ | 0.23 (0.26)* | 0.20 (0.24)* | 0.18 (0.23)* | - |
| $R_{free}$ | 0.26 (0.33)* | 0.24 (0.26)* | 0.24 (0.26)* | - |
| Complexes per asymmetric unit | 2 | 2 | 2 | - |
| Number of atoms | 78030 | 77878 | 36865 | - |
| Protein residues | 9989 | 9981 | 4661 | - |
| Ligand atoms | 496 | 496 | 248 | - |
| RMS bond lengths (Å) | 0.004 | 0.004 | 0.004 | - |
| RMS bond angles (°) | 0.94 | 0.98 | 0.93 | - |
| Ramachandran favored (%) | 94.0 | 94.0 | 95.0 | - |
| Ramachandran allowed (%) | 5.4 | 5.5 | 4.5 | - |
| Ramachandran outliers (%) | 0.3 | 0.3 | 0.2 | - |
| Clashscore | 4.5 | 4.8 | 5.6 | - |
| **Mean *B* values (Å$^2$)** | | | | |
| Overall | 108.4 | 98.3 | 48.6 | - |
| Macromolecules | 108.6 | 98.4 | 48.6 | - |
| Ligands | 78.5 | 91.5 | 36.4 | - |

*Numbers represent the highest-resolution shell.
†Numbers represent the truncated data after treated with ellipsoidal truncation and anisotropic scaling.
‡$R_{merge} = \Sigma_{hkl}\Sigma_i|I_i(hkl)-I_{av}(hkl)|/\Sigma_{hkl}\Sigma_iI_i(hkl)$.
DOI: https://doi.org/10.7554/eLife.38922.013

preceding TOG2, the remaining 40 residues of the linker were disordered (*Figure 2—figure supplement 1F–I*).

The dimension of each wheel-like assembly was 210 × 198 × 60 Å (*Figure 2A*). The 2:4:4 stoichiometry observed in the X-ray structure matched the stoichiometry measured for wt-Alp14-dimer:αβ-tubulin:DRP complexes (*Figure 1D*; *Figure 1—figure supplement 2D–F*). We hypothesized that sk-Alp14-monomer formed dimeric organization, despite the lack of dimerization domains, due to the high concentration of these complexes during crystallization. X-ray structures revealed two TOG1-TOG2 subunits in a pseudo-dimeric assembly forming the core of these complexes. In a TOG square, each TOG domain was bound to a curved αβ-tubulin capped by a DRP through its outward-facing binding interface and was not in contact with the neighboring TOG-bound αβ-tubulin (*Figure 2A*; *Figure 2—figure supplement 1F–H*). The distances and interaction patterns between residues of α-tubulin and DRP bound to a neighboring β-tubulin indicated that DRP only interacts with its cognate β-tubulin and does not bind a neighboring α-tubulin (*Figure 2—figure supplement 1J,K*). The latter suggests that DRP has no effect on stabilizing each TOG square assembly. Rather, DRP binding only caps β-tubulin, presenting a significant impediment to the polymerization of αβ-tubulins while bound to the TOG1-TOG2 subunits. The αβ-tubulins bound to the TOG square are positioned in a polarized orientation, due to the asymmetry in the TOG domain αβ-tubulin interface and pseudo-dimeric TOG1-TOG2 subunit interfaces within the TOG square (see below). The β-tubulin on a TOG1-bound αβ-tubulin is rotated approximately 90° from its polymer-forming interface relative to the adjacent α-tubulin on a TOG2-bound αβ-tubulin (*Figure 2B*).

## Two interfaces stabilize TOG1-TOG2 subunits into a TOG square assembly

The X-ray structures revealed that each TOG square is a dimer of TOG1-TOG2 array subunits assembled head-to-tail from alternating TOG1 and TOG2 domains. TOG domains were aligned along their narrow edges, analogous to four links attached head-to-tail forming an asymmetric square-like complex with two edges slightly longer than their orthogonal edges (*Figure 2C,D*). Two contact sites, which we term interfaces 1 and 2, stabilize the TOG square. These interfaces are formed by interactions formed via inter-HEAT repeat loops of each TOG domain, which are located on the opposite edges from the αβ-tubulin-binding sites. Although TOG1 and TOG2 domains are each 60 Å long, the TOG square assembly is slightly rectangular with 115 Å by 98 Å dimensions due to wider overlaps between TOG1 and TOG2 domains leading to 10 Å stagger at interface 1 sites, in contrast to the direct end-on corner-like interface 2 sites. Both interfaces 1 and 2 are stabilized by hydrophobic packing and ionic interaction zones (*Figure 2E–H*). Interface 1 packs a 668 Å$^2$ surface area and positions the TOG1 C-terminus at 90° to a 10-residue segment of the TOG1-TOG2 linker and the N-terminus of TOG2. The TOG1-TOG2 linker sequence forms an extended polypeptide that critically bridges interactions between TOG1 inter-HEAT repeat 6 α-helix/inter-HEAT 5–6 loop segment and the TOG2 inter-HEAT repeat 1-2/2-3 loop segments (*Figure 2E,F*). Interface 2 packs a 290 Å$^2$ surface area and positions the TOG2 C-terminus at 90° to the N-terminus of TOG1 (*Figure 2G,H*). In interface 2, the TOG2 inter-HEAT repeat 4–5 loop interacts with the TOG1 inter-HEAT repeat 1–2/HEAT1 α-helix (*Figure 2G*). The residues forming interface 1 and interface 2 within TOG1, TOG2, and linker regions are either moderately or highly conserved (*Figure 2F,H*; *Figure 2—figure supplement 2*). The total buried surface area stabilizing two sets of interfaces 1 and 2 in a TOG square is 1930 Å$^2$, which is moderate in size and dispersed for such a large assembly. This finding suggests that this conformation may be meta-stable and that DRP binding and its inhibition of αβ-tubulin polymerization may stabilize this intermediate.

## Cysteine crosslinking and mass spectrometry reveal Alp14-dimer forms TOG square assembly interfaces in solution

Next, we examined and chemically trapped the direct physical interactions between TOG1 and TOG2 interfaces observed in TOG square structures using cysteine mutagenesis and disulfide crosslinking. We generated mutants with specific cysteine pairs within the two sides of interface 1 (S180C, L304C) or interface 2 (S41C, E518C) in native dimeric sk-Alp14 (termed sk-Alp14-dimer; residues 1–724) (*Figure 3A,B*). We tested whether these interfaces formed inter-subunit contacts in dimeric TOG array by crosslinking via disulfide oxidation. A 110 kDa crosslinked species was observed in all

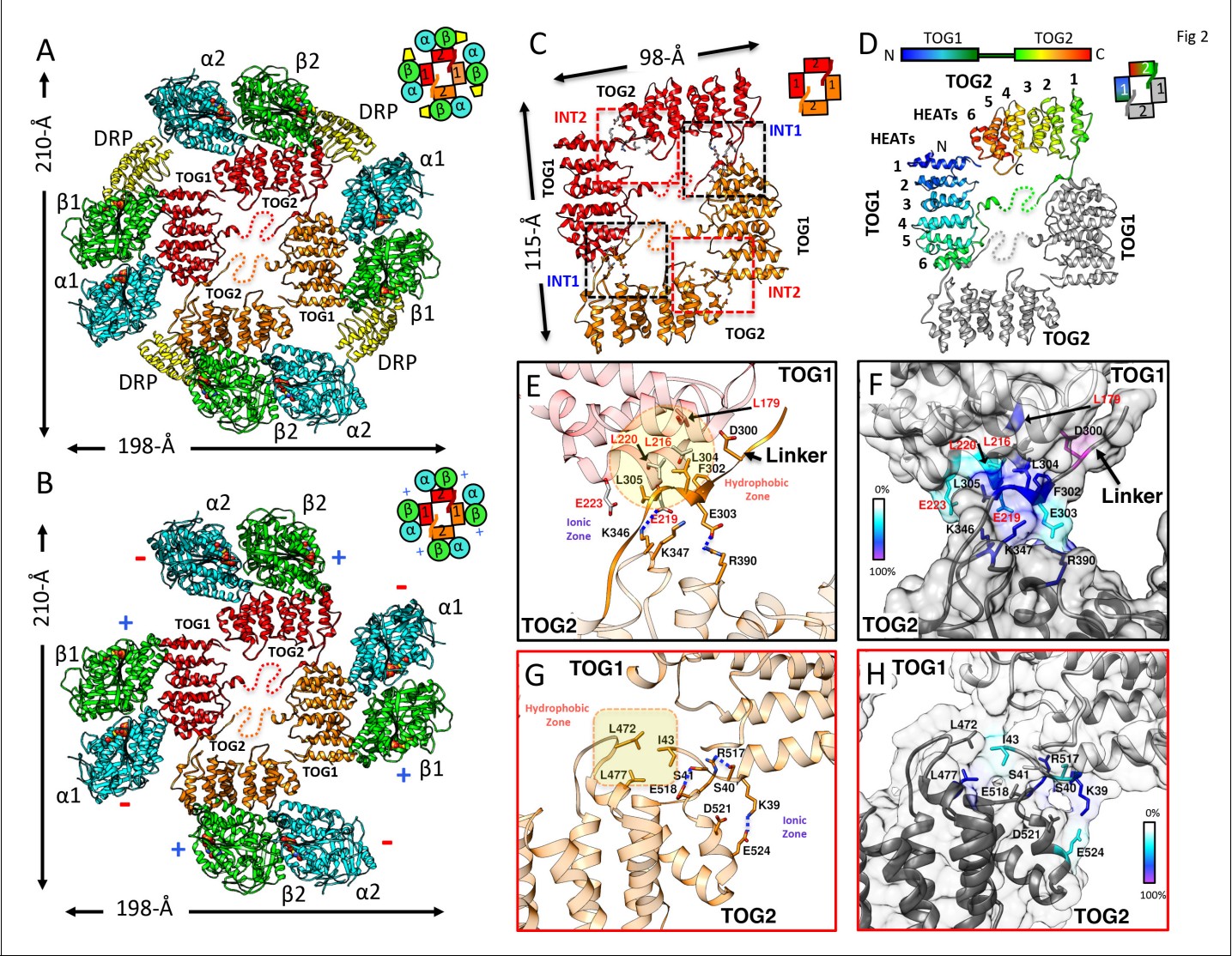

**Figure 2.** X-ray structures reveal αβ-tubulins bound in a wheel-like organization around a pseudo-dimeric TOG square complex. (A–B) 3.6 Å X-ray crystal structure of the *S. kluyveri* 1:2:2 sk-Alp14:αβ-tubulin:DRP reveals pseudo-dimeric head-to-tail subunits (red and orange) in a TOG square assembly consisting of four TOG domains bound to four αβ-tubulins (α-tubulin shown in cyan and β-tubulin shown in green) in a wheel-like organization. (A) Structure with DRP (yellow) bound to each αβ-tubulin. (B) Structure with DRP computationally removed. Each αβ-tubulin (α1β1) is positioned 90° rotated from its polymer-forming interface on its neighboring αβ-tubulin (α2β2). (C) Pseudo-dimeric TOG1-TOG2 subunits, shown in orange and red, respectively, form a head-to-tail TOG square (inset). Interface 1 is formed by the N-terminus of TOG2 and the TOG1-TOG2 linker binding to the C-terminus of the TOG1 domain of a second subunit, forming a 90° corner. Interface 2 is formed by the N-terminus of TOG1 binding the C-terminus of TOG2 within the same subunit in a 90° corner (*Figure 2—figure supplement 1I*). (D) Rainbow view of TOG1-TOG2 with N- and C-termini displayed in a blue-to-red color gradient, while the other subunit is displayed in grey. Each TOG is composed of six HEAT repeats (numbered). (E) Close-up view of interface 1. A hydrophobic zone stabilizes interface1 (yellow and highlighted by red outline) involving Leu220 (L220) and Leu217 (L217) of the TOG1 inter-HEAT 5–6 loop, Leu179 (L179) of the HEAT 6 A-helix in TOG1 (red ribbon) stabilized by linker residues (solid orange) Phe302 (F302), Leu304 (L304), and Leu305 (L305). An ionic zone guides interface 1 involving Glu219 (E219) of TOG1 inter-HEAT 5–6 loop and Glu305 (E305) of the TOG1-TOG2 linker, forming salt bridges with Lys346 (K346) and Lys347 (K347) of the TOG2 (light orange) inter-HEAT 1–2 loop and Arg390 (R390) of the TOG2 HEAT 2,3 loop, respectively. (F) Close-up view of interface 1, as in C, displaying residue conservation based on the alignment shown in *Figure 2—figure supplement 2*. (G) Close-up view of interface 2. A hydrophobic zone stabilizes interface 1 involving Leu477 (L477) and Leu472 (L472) of the TOG2 inter-HEAT4-5 loop with Ile43 of the TOG1 inter-HEAT1-2 loop. Ionic zone selectively guides interface 2, involving Lys39 (K39) and Ser41 (S41) of the TOG1 inter-HEAT1-2 loop and helix 1B with Arg517 (R517), Glu518 (E518), Asp521 (D521), and Glu524 (E524) of the TOG2 inter-HEAT5-6 loop and A-helix. (H) Close-up view of interface 2, as in D, displaying reside conservation based on the alignment in *Figure 2—figure supplement 2*.
DOI: https://doi.org/10.7554/eLife.38922.007

The following figure supplements are available for figure 2:

*Figure 2 continued on next page*

*Figure 2 continued*

**Figure supplement 1.** X-ray crystallography and structure determination of 2:4:4 Alp14-monomer:αβ-tubulin:DRP complexes.
DOI: https://doi.org/10.7554/eLife.38922.008
**Figure supplement 2.** Sequence conservation in TOG square interfaces across each TOG1 and TOG2 domain.
DOI: https://doi.org/10.7554/eLife.38922.009

conditions where soluble αβ-tubulin was added, and mass spectrometry (LC/MS-MS) confirmed that this intermediate was indeed a crosslinked α- and β-tubulin heterodimer (*Figure 3C,D*; *Figure 3—figure supplement 1A*). We also observed a ~ 170 kDa species that specifically formed in the αβ-tubulin-bound sk-Alp14 S180C-L304C mutant and not in the native sk-Alp14-dimer or the sk-Alp14-S41-E518C mutants. Furthermore, this 170 kDa intermediate was also not observed with sk-Alp14-S180C-L304C without αβ-tubulin (*Figure 3D*). Mass spectrometry confirmed that this 170 kDa intermediate was indeed the sk-Alp14-S180C-L304C protein. Next, we mapped the cysteine residues involved in disulfide bonds in sk-Alp14-S180C-L304C mutants through peptide disulfide mapping after differential alkylation and mass spectrometry (see Materials and methods). This approach revealed only two classes of peptides in sk-Alp14-S180C-L304C with 105 Da of mass added onto the cysteines, suggesting that they were engaged in disulfide bonds with the following sequence boundaries: 297–320 and 179–189 (*Figure 3—figure supplement 1B*). These two peptide sequences represent TOG1 inter-HEAT-repeat and TOG1-TOG2 linker regions, both of which are involved in forming interface 1 in the X-ray structures (*Figure 2*). All the remaining peptides with cysteine residues that were identified in sk-Alp14-S180C-L304C included 57 Da in added mass, suggesting that they were in the reduced form and did not form disulfide bridges. Thus, these data directly provide independent support of interface 1 of the TOG square conformation forming in sk-Alp14-dimer in solution outside of the crystallographic setting, and of indeed being the inter-subunit dimeric interface between two TOG-array subunits, as visualized in the crystal structures (*Figure 3A,B*).

## Disrupting TOG square assembly interfaces destabilizes organization, but does not affect αβ-tubulin binding

We explored the role of interfaces 1 and 2 in stabilizing TOG square assembly and their effect on the αβ-tubulin capacity of TOG arrays. We generated three Alp14-dimer mutants that harbored either partially or fully disrupted interface 1 and 2 sites (*Figure 4A–D*). We targeted disruption of salt bridges or hydrophobic zones in interfaces 1 and 2 by mutating conserved alanines, leucines, or glutamates (*Figure 2E,G*; *Figure 4A–D*). Charged residues were either replaced with alanines or residues of the opposite charge, and hydrophobic residues were replaced with charged residues to dissociate hydrophobic interactions (*Figure 4A–D*). Initially, we disrupted interfaces 1 and 2 using one-, two-, or three-residue mutations in wt-Alp14-dimer. However, these mutants showed substantial levels of TOG square assemblies as assessed by negative stain electron microscopy (EM) (data not shown). Thus, we aimed to fully disrupt interfaces 1 and 2 by using seven to eight residues per interface. We mutated 8 residues in wt-Alp14-dimer to disrupt interface 1 (termed INT1; *Figure 4B*), 7 residues in wt-Alp14-dimer to disrupt interface 2 (termed INT2; *Figure 4C*), or 15 residues to disrupt both interfaces 1 and 2 (termed INT1 +2; *Figure 4D*) (see Materials and methods).

We studied the αβ-tubulin-binding capacities and stoichiometries of INT1, INT2, and INT1 +2 compared to wt-Alp14-dimer using quantitative-SEC and SEC-MALS approaches, respectively, as described in *Figure 1*. INT1, INT2, and INT1 +2 mutants bound nearly identical quantities of αβ-tubulin to wt-Alp14-dimer (*Figure 4E*; *Figure 4F,G,H*; *Figure 4—figure supplement 1*). INT1, INT2, and INT1 +2 bound approximately four αβ-tubulins at 80 mM KCl as assessed by SEC-MALS (*Tables 1* and *2*; *Figure 4F,G,H*) and approximately half of the bound αβ-tubulin dissociated at 200 mM KCl as quantitated by quantitative-SEC (*Figure 4D*; *Figure 4—figure supplement 1D,E,F*).

We next used negative stain EM and 2D-single particle image analyses to compare the conformations of four αβ-tubulin-bound wt-Alp14-dimer assemblies to the αβ-tubulin:INT1, INT2 and INT1 +2 mutant assemblies. Negative stain images showed that 4:2 wt-Alp14-dimer:αβ-tubulin complexes formed two types compact particle-like assemblies of either 15 or 19 nm diameter compact circular complexes (*Figure 2*) matched the general features previously described for yeast Stu2p-tubulin or XMAP215-tubulin we previously described (*Al-Bassam et al., 2006*; *Brouhard et al., 2008*). 2D-image class averages were compared via projection-matching to low resolution-filtered structural

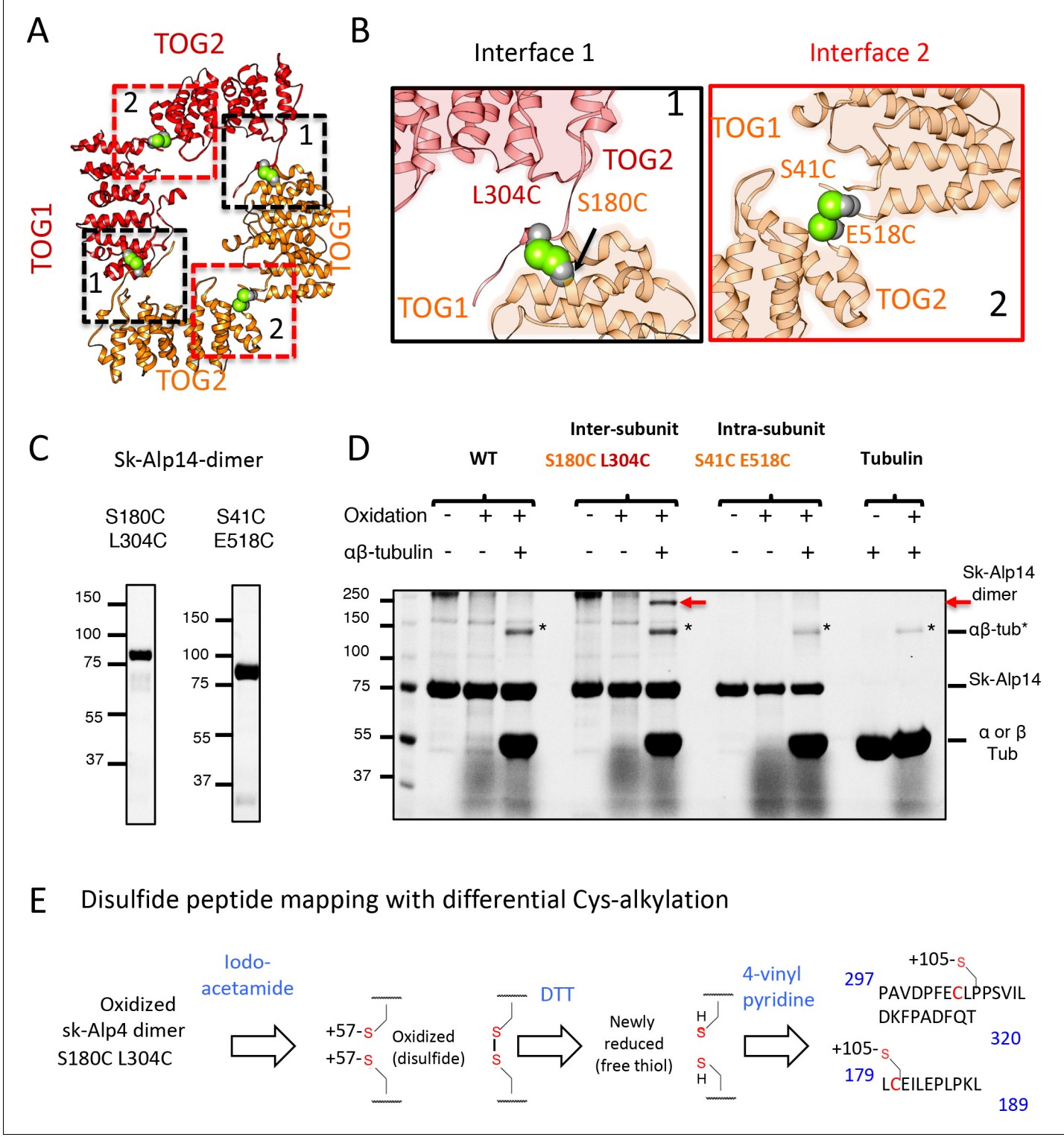

**Figure 3.** Cysteine mutagenesis/crosslinking and mass spectrometry-based peptide-mapping reveal that dimeric sk-Alp14 forms TOG square conformations in solution. (**A**) TOG square structure showing two cysteine pairs (green space fill) mutated in interfaces 1 (black-dashed lines) and 2 (red-dashed lines). (**B**) Close-up views of interfaces 1 (left) and 2 (right) showing the S180C-L304C and E518C-S41C residue pairs, respectively (green space fill), in sk-Alp14. (**C**) SDS-PAGE of SEC-purified sk-Alp14 S180C-L304C and E518C-S41C. (**D**) Crosslinking studies of cysteine structural-based mutants using oxidizing conditions and αβ-tubulin binding, as denoted by (+) and (–). The αβ-tubulins form an intermediate in oxidizing conditions observed in all conditions that include αβ-tubulin (marked '*'). S180C-L304C sk-Alp14 forms a dimeric 170 kDa intermediate upon oxidization and αβ-tubulin binding (red arrow). In contrast, wt-Alp14 or sk-Alp14 E518C-S41C do not form this intermediate. (**E**) Disulfide peptide mapping of cysteines in sk-Alp14

*Figure 3 continued on next page*

*Figure 3 continued*

S180C-L304C using differential alkylation and LC/MS-MS. We used mass spectrometry (LC/MS-MS) after a differential alkylation strategy (*Figure 3—figure supplement 1D*) to map peptides with disulfides. Briefly, oxidized sk-Alp14 S180C-L304C (170 kDa) SDS-PAGE-purified bands were subjected to proteolysis and treated with iodoacetamide. Under these conditions, 57 Da in mass is added to peptides with reduced cysteines (free thiols), without affecting disulfides. Dithiothreitol was then used to reduce peptides with disulfides, which were then treated with 4-vinyl pyridine, which added 105 Da in mass to peptides with newly formed free thiols. Using LC/MS-MS, peptides with modified cysteines were identified based on added mass. Details provided in *Figure 3—figure supplement 1D*.
DOI: https://doi.org/10.7554/eLife.38922.011
The following figure supplement is available for figure 3:

**Figure supplement 1.** Yeast dimeric TOG arrays form a TOG square assembly in solution as measured by cysteine crosslinking and mass spectrometry.
DOI: https://doi.org/10.7554/eLife.38922.012

models for a TOG square with and without four αβ-tubulins, revealing TOG square bound to four αβ-tubulins matched well the density organization of the 19 nm diameter 2D-class averages (*Figure 4I*; *Figure 4—figure supplement 2B*). 2D-projections of a low-resolution filtered TOG square model without αβ-tubulins bound matched well the organization of the 15 nm diameter class averages suggesting those classes represent TOG squares that likely lost their bound αβ-tubulins on the grid (*Figure 4I*; *Figure 4—figure supplement 2C*). These data provide another line of support that TOG array subunits form square assemblies in solution, and that they match the organization of a TOG square bound to four αβ-tubulins (termed wheels) as observed in the crystal structure (*Figure 2*) or these dissociated from αβ-tubulin leading to isolated square-shaped assemblies (termed squares) (*Figure 4*). In contrast, INT1:αβ-tubulin complexes did not form square assemblies, and particles exhibited either open flexible organization with many inter-connected 8 nm long inter-connected densities or 16 nm long short filaments. 2D-image classification showed either 16 nm curved filaments, two 8 nm densities at right angles, or isolated 8 nm densities. 2D-projection matching using low resolution filtered models for single TOG1-TOG2 subunit bound to two αβ-tubulins from a TOG square structure, polymerized TOG-αβ-tubulin complexes (see next section), or with individual TOG-αβ-tubulin complexes (*Ayaz et al., 2012*) matched well to the three types of class averages (*Figure 4J*; *Figure 4—figure supplement 2D–F*). These data confirmed that these complexes were indeed either single TOG1-TOG2 subunits, with 90 degree pre-arranged and non-polymerized αβ-tubulin assemblies, TOG1-TOG2 bound to two polymerized αβ-tubulins assemblies, or disordered assemblies composed of isolated TOG-αβ-tubulin complexes. INT2-αβ-tubulin complexes showed similar pattern of class averages that matched similar models as INT1-αβ-tubulin complexes suggesting similarly dissociation of a single interface in the TOG square (*Figure 4K*; *Figure 4—figure supplement 2G–I*). While, INT1 +2 αβ-tubulin complexes showed only dissociated, flexibly connected necklaces of 8 nm densities of αβ-tubulin (*Figure 4L*). 2D-image classification of these particles and 2D-projection matching of these complexes showed that each 8 nm class averages matched 2D-projections of a TOG-bound αβ-tubulin (*Figure 4L*; *Figure 4—figure supplement 2J–K*). Our biochemical and negative stain-EM analyses suggest that wt-Alp14 dimer TOG arrays form square-shaped assemblies that match the organization observed in the TOG square crystal structure. Specific aspects of TOG square organization are clearly disrupted leading to the predicted defects in organization in the INT1, INT2, and INT1 +2 mutants without any effect on αβ-tubulin binding (*Figure 4J, K,L*). An interesting observation is that INT1 and INT2 mutants showed the propensity to form two polymerized αβ-tubulins filaments in some cases, suggesting that spontaneous in-solution αβ-tubulin polymerization occurs in the case of interface 1 and interface 2 destabilization (*Figure 4J,K*; *Figure 4—figure supplement 2E,H*). The dual inactivation of both TOG square interfaces in INT1 +2 resulted in dissociated TOG-αβ-tubulin complexes with a poor ability to form polymerize αβ-tubulins (*Figure 4L*; *Figure 4—figure supplement 2*).

## X-ray structure of a polymerization complex: TOG1-TOG2 subunit unfurling promotes the concerted polymerization of two αβ-tubulin

The TOG square conformation shows how αβ-tubulins are recruited to TOG arrays but does not reveal how TOG arrays drive polymerization the recruited αβ-tubulins. We hypothesized that the TOG square structure may undergo a subsequent conformational change to promote polymerization of the recruited αβ-tubulins. To explore this transition, we created a biochemical approach to

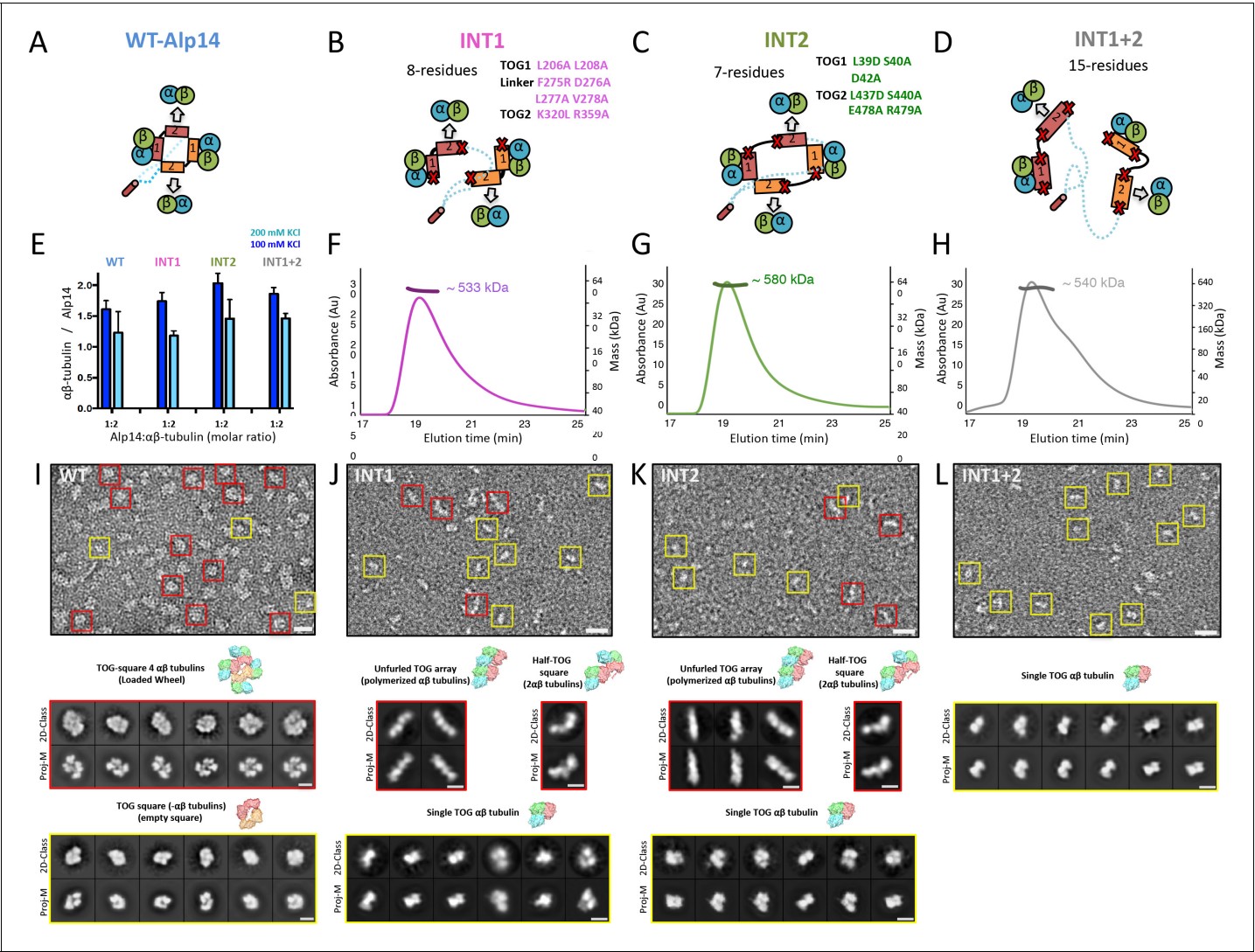

**Figure 4.** Inactivation of interfaces 1 and 2 destabilizes TOG square organization without affecting αβ-tubulin binding. (A–D) Generation of structure-based TOG square assembly-defective mutants using wt-Alp14-dimer (A) through inactivation of interface 1 in the INT1 mutant (B: INT1, pink; eight mutant residues), interface 2 in the INT2 mutant (C: INT2, green; seven mutant residues), or both interfaces 1 and 2 in the INT1 +2 mutant (D: INT1 +2, grey; 15 mutant residues). (E) Summary of SEC-measured αβ-tubulin-binding molar ratios of INT1, INT2, and INT1 +2 compared to wt-Alp14-dimer as described for *Figure 1D* suggests no defects in αβ-tubulin binding at 100 mM KCl (blue) and a similar decrease in αβ-tubulin binding upon 200 mM KCl ionic strength increase (cyan). Additional information is described in *Figure 4—figure supplement 1*. (F–H) SEC-MALS of INT1:αβ-tubulin (F), INT2:αβ-tubulin (G), and INT1 +2: αβ-tubulin (H) complexes at 2:4 stoichiometry at 100 mM KCl (masses reported in *Table 1*). SEC-MALS reveals similar mass to wt-Alp14-dimer complexes with αβ-tubulin which are reported *Table 2*. (I) Top, raw negative stain EM image of wt-Alp14-dimer:αβ-tubulin at 100 mM KCl reveals wheel-shaped assemblies that are 15–19 nm in diameter as previously described for Stu2-αβ-tubulin complexes (*Al-Bassam et al., 2006*). Middle panel, 2D-classes reveal 19 nm wheel-shaped particles that match the 2D-projection of 30 Å resolution-filtered 2:4 TOG-square:αβ-tubulin complex (shown above the panel). These classes match the organization observed in the structure described in *Figure 2*. Bottom panel, second group of 2D-classes reveal small diamond-shaped particles that match the 2D-projection of a 30 Å resolution-filtered model of the TOG square without αβ-tubulins (shown above the panel). (J) raw image of INT1:αβ-tubulin reveals elongated conformations with either bent-conformations composed of bent-particles with two 8 nm densities at 90-degree angles or 16 nm filament-like particles. Middle panel, 2D-classes reveal 16 nm elongated classes that match the 2D-projection of either 30 Å resolution-filtered TOG1-TOG2 bound to two polymerized tubulins (shown above the second row) as described structurally in the next section. Left panel, 90-degree bend class averages that match 2D-projections of a resolution-filtered model of one TOG1-TOG2 subunit of the TOG square bound to two tubulins at 90 degrees (as shown above). Bottom panel, 2D-classes reveal 8 nm lengths, which match 2D-projections of low-resolution-filtered TOG-αβ-tubulin complexes (PDB ID: 4FFB shown above the panel). (K) INT2 forms extended necklace shaped or extended 16 nm minifilaments. Middle panel, 2D-class averages reveal either 16 nm filament-like particles match 2D-projections of TOG1-TOG2 bound to two polymerized αβ-tubulins (left) or bent particles that match 2D-projections of a bend TOG1-TOG2 subunit bound to two non-polymerized αβ-tubulins, as observed in half a TOG square complex (right). (L) INT1 +2:αβ-tubulin complexes form only dissociated assemblies with

*Figure 4 continued on next page*

*Figure 4 continued*

randomly interconnected 8 nm assemblies. Bottom panel, 2D-classes reveal 8 nm lengths, which match 2D-projections of low-resolution-filtered TOG-αβ-tubulin complexes (PDB ID:4FFB shown above the panel). Additional information can be found in *Figure 4—figure supplement 1*.

DOI: https://doi.org/10.7554/eLife.38922.014

The following figure supplements are available for figure 4:

**Figure supplement 1.** Inactivating interfaces 1 and 2 destabilizes the TOG square organization but does not influence αβ-tubulin-binding activity.

DOI: https://doi.org/10.7554/eLife.38922.015

**Figure supplement 2.** Negative stain EM micrographs and class averages for the TOG square and its inactivated assemblies.

DOI: https://doi.org/10.7554/eLife.38922.016

partially release αβ-tubulin from polymerization by relieving DRP inhibition of αβ-tubulin polymerization while they are bound to TOG arrays. We reasoned that a structural transition may occur more readily if DRP dissociates from β-tubulin in a crystallization setting. We developed a strategy to conditionally release αβ-tubulin from polymerization while being recruited into TOG arrays by using a weakened affinity DRP. We reasoned that the increased dissociation of DRP may allow complexes to form polymerized αβ-tubulin intermediates in steady state, as seen in the negative stain studies. To accomplish this, we removed the N-terminal ankyrin repeat of DRP (herein termed DRPΔN). We measured DRPΔN affinity using ITC, revealing a three-fold decrease in its αβ-tubulin-binding affinity as compared to DRP (*Figure 5A–B*). During purification, complexes of 1:2:2 sk-Alp14-monomer:αβ-tubulin:DRPΔN behaved similarly to those assembled with DRP on SEC (*Figure 5C*; *Figure 5—figure supplement 1*). However, using crystallization conditions identical to those used to obtain the 4:4:2 TOG square conformation, we obtained crystals that exhibited a distinct rectangular morphology using sk-Alp14-monomer or sk-Alp14-dimer αβ-tubulin:DRPΔN complexes (see Materials and methods; *Figure 5—figure supplement 2A*). These crystals formed in the same space group (P2$_1$) with distinct unit cell dimensions compared to the TOG square crystals (*Table 3*). These crystals exclusively formed only when DRPΔN was used with αβ-tubulin:sk-Alp14-monomer or -dimer complexes. We determined an X-ray structure to 3.2 Å resolution by molecular replacement using these crystals (*Figure 5—figure supplement 2B*). The structure revealed a novel assembly consisting of complexes with the stoichiometry of 1:2 sk-Alp14-monomer:αβ-tubulin and a single DRPΔN positioned on the top end of TOG2-bound β-tubulin (termed the 1:2:1 structure; *Figure 5E,F*; *Figure 5—figure supplement 2C–D*).

The refined 3.2 Å 1:2:1 X-ray structure revealed an extended conformation with two αβ-tubulins polymerized head-to-tail in a curved protofilament (*Figure 5E–F*). In this conformation, TOG1 and TOG2 did not form any interactions with each other and their adjoining linker became disordered (*Figure 5E,F*; *Figure 5—figure supplement 2C,D*). TOG1 and TOG2 were specifically bound to the lower and upper αβ-tubulins, respectively, of a highly curved protofilament. Only a single DRPΔN capped the TOG2-bound αβ-tubulin (*Figure 5E*). Compared to the TOG square, this 'unfurling' rearrangement required 68°-rotation and 32 Å translation of TOG2:αβ-tubulin hinging around interface 2 and TOG1-αβ-tubulin (*Figure 6A,B*). This transition promoted the concerted polymerization of TOG2:αβ-tubulin onto the plus-end of TOG1:αβ-tubulin, and consequently drove the dissociation of the second DRPΔN (*Figure 5C,D*). The two αβ-tubulin polymers in this complex were highly curved protofilaments (16.4° inter-dimer interface). This protofilament structure displayed ~3° more curvature than RB3/stathmin-αβ-tubulin curved protofilament structures (*Table 4*; *Figure 5—figure supplement 2E*) (*Nawrotek et al., 2011*). Comparison of the αβ-tubulin dimer structure (α2β2) within this 1:2:1 structure to the unpolymerized αβ-tubulin structure revealed that polymerization is associated with a 5° rotation and 10 Å translation in the T7 loop and H8 helix of the TOG2-bound α-tubulin, which engages TOG1-bound β-tubulin elements and the E-site-bound GDP nucleotide (*Figure 6C,D*). The latter conformational change likely stabilizes inter-dimer αβ-tubulin interfaces (*Figure 5D*), burying a 1650 Å inter-dimer interface (*Figure 5E,F*). This conformational change occurs at a site similar to those observed during the MT lattice GTP hydrolysis transition (*Alushin et al., 2014*). Thus, the 1:2:1 unfurled structure represents a concerted αβ-tubulin post-polymerization intermediate promoted by a single TOG1-TOG2 subunit prior to straightening the protofilament during polymerization at MT plus-ends.

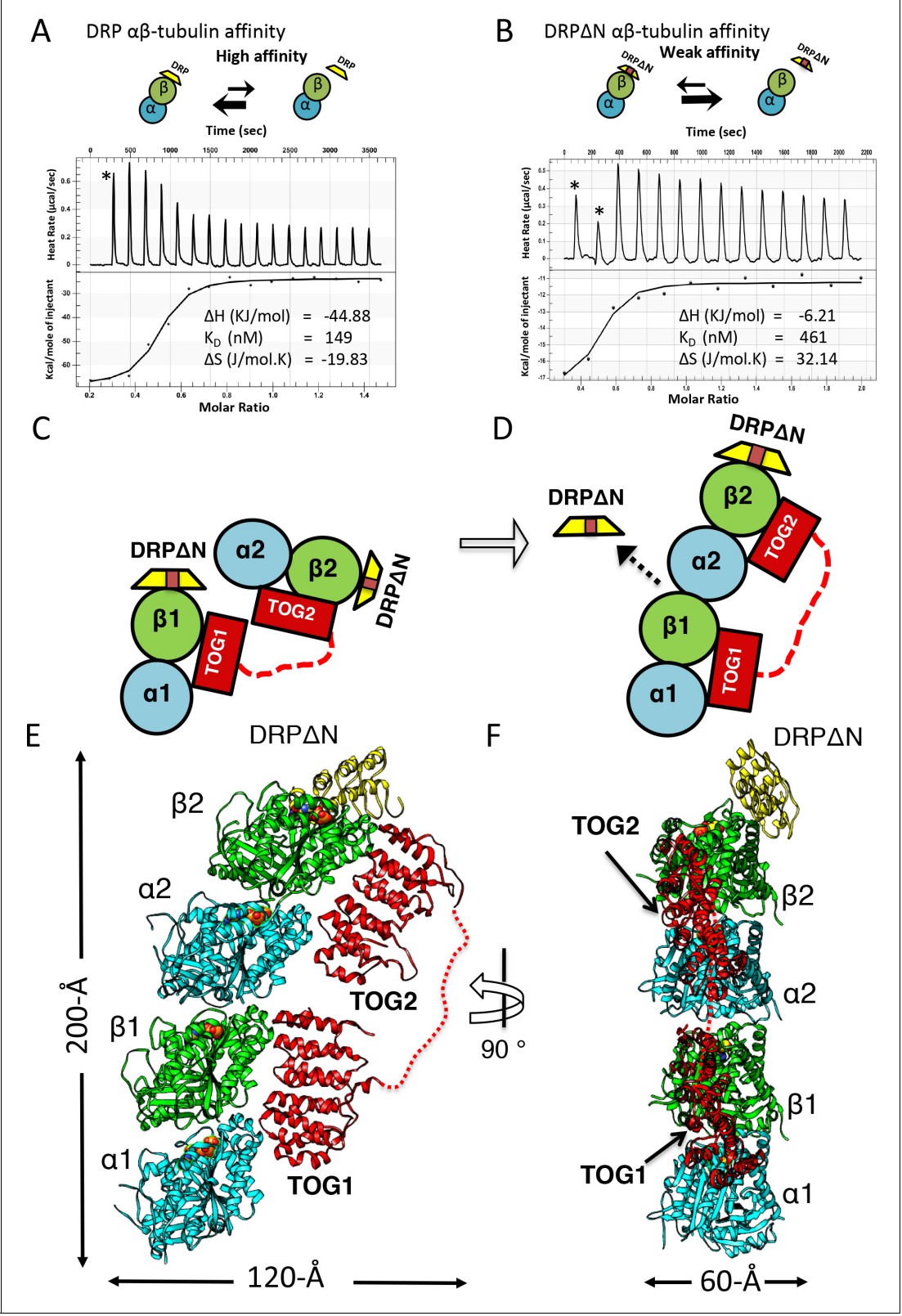

**Figure 5.** X-ray structure of 1:2:1 TOG-array:αβ-tubulin:DRPΔN reveals unfurled TOG1-TOG2 array bound to two polymerized αβ-tubulins. (A, B) Top, schemes of DRP and DRPΔN binding to αβ-tubulin. DRP shifts the equilibrium toward dissociation from αβ-tubulin. Bottom, isothermal titration calorimetery studies reveal a three-fold decrease in affinity of DRPΔN binding to αβ-tubulin (461 nM) compared to DRP (149 nM). (C, D) Two schematic views of the TOG1-TOG2 αβ-tubulin complex transition from the TOG square (only half is shown) to the unfurled conformation upon DRPΔN

*Figure 5 continued on next page*

*Figure 5 continued*

dissociation. (E, F) Two orthogonal views of the 3.2 Å X-ray structure of 1:2:1 sk-Alp14 (red):αβ-tubulins (cyan and green):DRPΔN (yellow) complex, indicating a polymerized protofilament state. TOG2 and TOG1 are bound to the upper (α2β2) tubulin and lower (α1β1) tubulin, respectively.

DOI: https://doi.org/10.7554/eLife.38922.018

The following figure supplements are available for figure 5:

**Figure supplement 1.** Strategy to promote αβ-tubulin polymerization using DRPΔN and the structural comparison of DRP and DRPΔN interfaces with αβ-tubulin.

DOI: https://doi.org/10.7554/eLife.38922.019

**Figure supplement 2.** X-ray crystallographic structure determination of 1:2:1 Sk-Alp14-monomer:αβ-tubulin:DRPΔN complex.

DOI: https://doi.org/10.7554/eLife.38922.020

## Modeling αβ-tubulin-bound TOG square and unfurled structure docking onto microtubule plus-ends

We next evaluated how X-ray structures of αβ-tubulin-bound TOG squares and unfurled TOG1-TOG2 αβ-tubulin assemblies can dock onto protofilament tips at MT plus-ends. Atomic models for these states were overlaid onto the terminal αβ-tubulins of curved GTP or GDP αβ-tubulin protofilament models (*Figure 7*). Attempts to dock the αβ-tubulin onto protofilament ends exposed at the MT minus-end caused substantial steric clashes, supporting the notion that TOG square states are completely incompatible with docking at MT minus-ends (data not shown). The four αβ-tubulin-bound TOG square assembly X-ray structure (*Figure 2*) was superimposed onto that of the terminal αβ-tubulin at protofilament ends in two docking orientations, either via the αβ-tubulins bound onto TOG1 or TOG2 (*Figure 7A,B*). We observed a slight steric surface overlap between the four αβ-tubulin-loaded TOG square and the curved protofilament when TOG1-αβ-tubulin was docked onto the β-tubulin at the protofilament plus-end (*Figure 7A*). This steric overlap was caused by overlap between αβ-tubulin-TOG2 from the second TOG1-TOG2 subunit in the TOG square with penultimate αβ-tubulin from the protofilament end (*Figure 7A*; *Figure 7—figure supplement 1A*). In contrast, we observed no steric contact when the TOG square was docked via αβ-tubulin-TOG2. In this orientation, the TOG1-αβ-tubulin from the second subunit was retracted by 10 Å from the penultimate αβ-tubulin in the protofilament in contrast to the TOG1-αβ-tubulin docking (*Figure 7B*). The differences between steric overlap of the TOG square with the protofilament in these two docking orientations were due to the asymmetric dimensions of the TOG square, caused by stagger between TOG1 and TOG2 domains at interface 1 compared to interface 2. These differences suggest that the destabilization of the TOG squares is more likely if TOG1-αβ-tubulin docks onto the protofilament plus-end in contrast to TOG2-αβ-tubulin docking. The unfurled 1:2 TOG1-TOG2:αβ-tubulin assembly can only be docked using TOG1-αβ-tubulin onto the protofilament plus-end and suggests that TOG2:αβ-tubulin is positioned the furthest away from the MT plus-end in this conformation. These models were used to assemble steps for a new MT polymerase model described in the discussion (*Figure 8*).

## Discussion

### A 'polarized unfurling' model for TOG arrays as MT polymerases

Using the combination of structural and biochemical analyses, we propose a new model for TOG arrays as MT polymerases, which we term the 'polarized-unfurling' model. The model is summarized in *Figure 8* and animated in *Video 1*. This model is supported by X-ray structures of two states, negative EM studies of αβ-tubulin complexes of wt-Alp14-dimer and three interface inactivated Alp14 mutants, differences in affinities of TOG1 and TOG2 domains for αβ-tubulins, and described via docking of models at protofilament ends at each step (*Figure 7*). Our model suggests that TOG arrays form two separate conformations that together promote MT polymerase activity: an αβ-tubulin recruitment complex as denoted by the TOG square (*Figures 2–4*) and an unfurled MT plus-end polymerization complex denoted by the polymerized 1:2:1 TOG1-TOG2: αβ-tubulin X-ray structures (*Figure 5*). We have effectively trapped these states by regulating the polymerization propensities for αβ-tubulins using DRP affinity while bound to TOG arrays (*Figures 2* and *5*). We hypothesize that the association of the αβ-tubulin-bound TOG square onto the MT plus-ends, via β-tubulin binding,

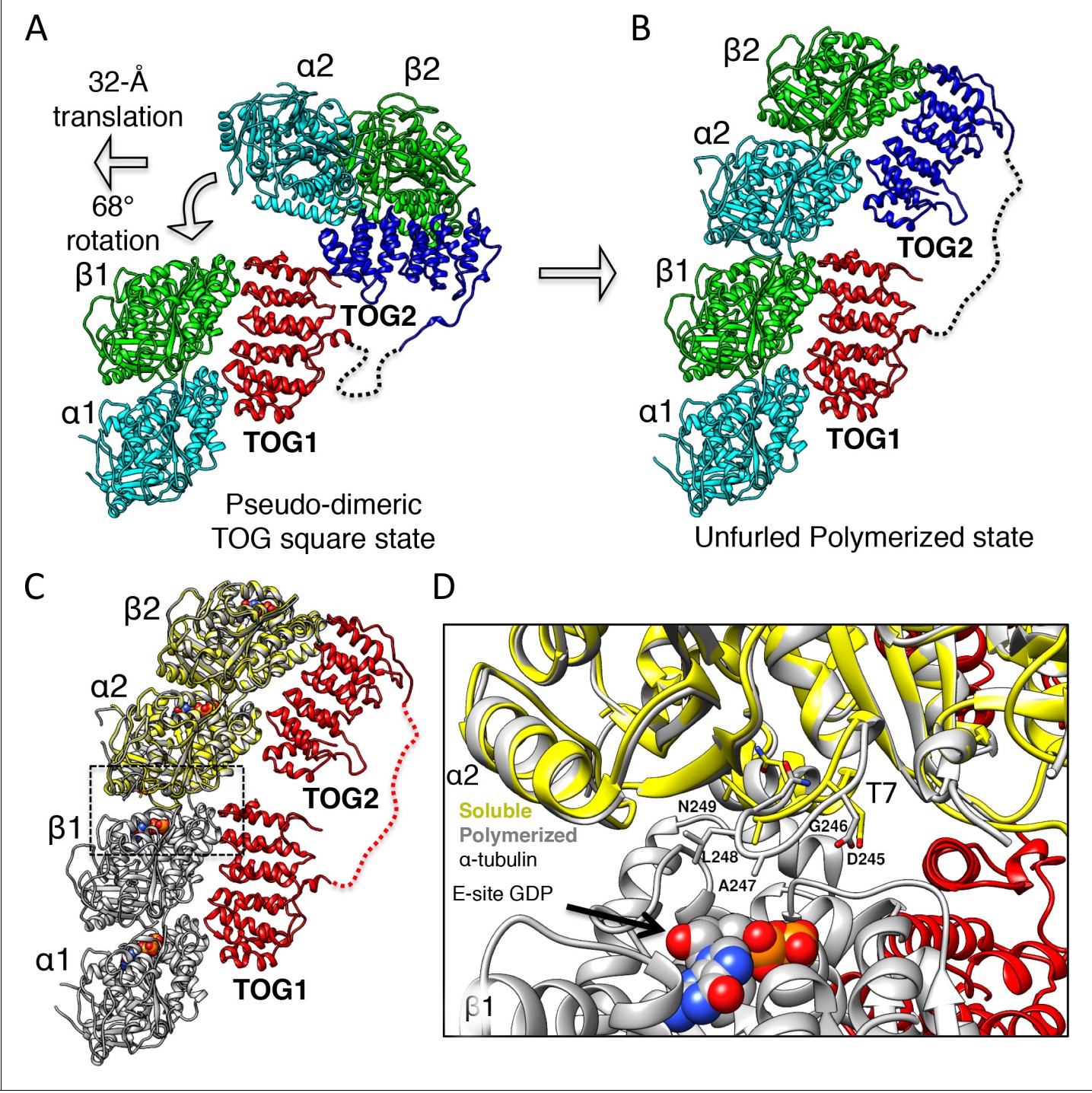

**Figure 6.** Unfurling the TOG array: TOG2 rotation around TOG1 promotes the bound αβ-tubulins to polymerize. (**A and B**) Conformational change of TOG2 (blue) around TOG1 (red) while each is bound to αβ-tubulin (green and cyan) from a corner subunit in the wheel assembly (left) and in the extended conformation (right). TOG2 rotates 32° and translates 68 Å upon release to drive αβ-tubulin polymerization into a protofilament. (**C**) Superimposing unpolymerized αβ-tubulin (yellow) onto the α2β2-tubulin shows a conformational change in α-tubulin at the inter-dimer interface induced by polymerization. (**D**) Close-up view of the polymerized inter-dimer interface. Unpolymerized αβ-tubulin (yellow) is superimposed onto α2β2 (grey) of 1:2:1 structure. The α2-tubulin T7 loop and H8 helix engage the β1-tubulin GDP nucleotide through a T7 loop 5 Å translation and H8 helix 5° rotation involving residues Asp245 (D245), Gly246 (G246), Ala247 (A247), and Leu248 (L248).

DOI: https://doi.org/10.7554/eLife.38922.022

**Table 4.** Buried Surface Area between αβ-tubulin dimer and TOG domains or DRP.

| Interface | 1:2:2: sk-Alp14-monomer: αβ-Tubulin: DRP(Å²)* | 1:2:2: sk-Alp14-monomer-SL: αβ-Tubulin: DRP (Å²) | 1:2:1: sk-Alp14-monomer: αβ-Tubulin: DRP-ΔN (Å²) |
|---|---|---|---|
| TOG1 and TOG2-interface 2 | 273 ± 35 | 257 ± 42 | - |
| TOG1-TOG2 dimer-interface 1 | 661 ± 27 | 702 ± 18 | - |
| αβ-tubulin and TOG1 | 752 ± 12 | 786 ± 42 | 804 ± 24 |
| αβ-tubulin and TOG2 | 916 ± 23 | 890 ± 22 | 863 ± 12 |
| β-tubulin and DRP or DRP-ΔN | 842 ± 68 | 873 ± 21 | 846 ± 20 |
| α-tubulin and DRP (inter-αβ-tubulin subunit) | 108 ± 47 | 81 ± 31 | - |

*Interface areas were determined by a single buried surface, and averaged among each non-crystallographic unit in the structure.
DOI: https://doi.org/10.7554/eLife.38922.017

drives the destabilization of the TOG square state and promotes the concerted unfurling of the TOG-αβ-tubulins to polymerize into a new protofilament.

## The polarized unfurling MT polymerase cycle

We envision the polarized unfurling model as follow: 1) Upon recruiting four soluble αβ-tubulins, dimeric TOG1-TOG2 arrays in proteins such Alp14 or Stu2, organize into compact TOG square assemblies in solution (*Figure 8A*). These assemblies place αβ-tubulins in close proximity with each other in a near head-to-tail polarized while inhibiting spontaneous polymerization. This 'ready-to-polymerize' orientation is due to the asymmetry in each TOG domain:αβ-tubulin interface and the unique head-to-tail assembly formed by two TOG1-TOG2 array subunits formed in the TOG square (*Figure 2*). 2) The αβ-tubulin-loaded TOG square assemblies diffuse along the MT lattice, mediated by an interaction of the SK-rich regions immediately C-terminal to TOG2, with acidic tubulin C-termini exposed on MT surfaces (*Figure 8A*). Proximity of the SK-rich region to the TOG array is essential for MT polymerase activity, and increasing its polypeptide distance causes MT polymerase defects (*Geyer et al., 2018*; *Al-Bassam et al., 2012*; *Brouhard et al., 2008*). 3) When the αβ-tubulin-loaded TOG squares reach the β-tubulins exposed at MT plus-end protofilament tips, α-tubulin of the TOG1- or TOG2-bound αβ-tubulin may polymerize with β-tubulin exposed at the MT plus-end tip (*Figure 8B–I*; *Figure 2B*). Although TOG squares may diffuse to MT minus-ends, docking onto α-tubulin at MT minus-ends is highly sterically inhibited, precluding the possibility of αβ-tubulin TOG square docking onto MT minus-ends. The αβ-tubulin docking of either TOG1 or TOG2 onto β-tubulin at MT plus-ends is likely to be random. However, two features of TOG domains favor TOG1-αβ-tubulin over TOG2 αβ-tubulin in docking onto MT plus-ends: A) TOG1 is more likely to be fully occupied by αβ-tubulin due to its slow exchange and high affinity compared to the rapid exchange of αβ-tubulin onto TOG2 (*Figure 1*). B) Steric overlap with MT protofilaments develops only if the TOG square docks via TOG1-αβ-tubulin but not if TOG2-αβ-tubulin docks (*Figure 7*; *Figure 8A–I and B–I*; *Figure 7—figure supplement 1A*). Thus, TOG squares docking to protofilament ends via TOG1-

**Table 5.** Intra and inter dimer curvature angles (°) of αβ-Tubulins observed in structures.

| | PDB ID | Intra dimer (α1β1) angle (°) | Inter dimer (β1α2) angle (°) |
|---|---|---|---|
| Stathmin:RB3 complex with GTP | 3RYH | 9.2 | 10.3 |
| Stathmin:RB3 complex with GDP | 1SA0 | 13.0 | 13.5 |
| αβ-tubulin:TOG1 complex with GTP | 4FFB | 13 | - |
| αβ-Tubulin:TOG2 complex with GTP | 4U3J | 13 | - |
| αβ-Tubulin:DRP complex with GDP | 4DRX | 11.0 | - |
| Sk-Alp14-monomer:αβ-Tubulin:DRP wheel with GDP | Current | 11.3 | - |
| Sk-Alp14-monomer:αβ-Tubulin:DRP-ΔN with GDP | Current | 11.2 | 16.4 |

*Curvature of αβ-tubulin interface were determined as previously described by Rice and Brouhard 2015.
DOI: https://doi.org/10.7554/eLife.38922.021

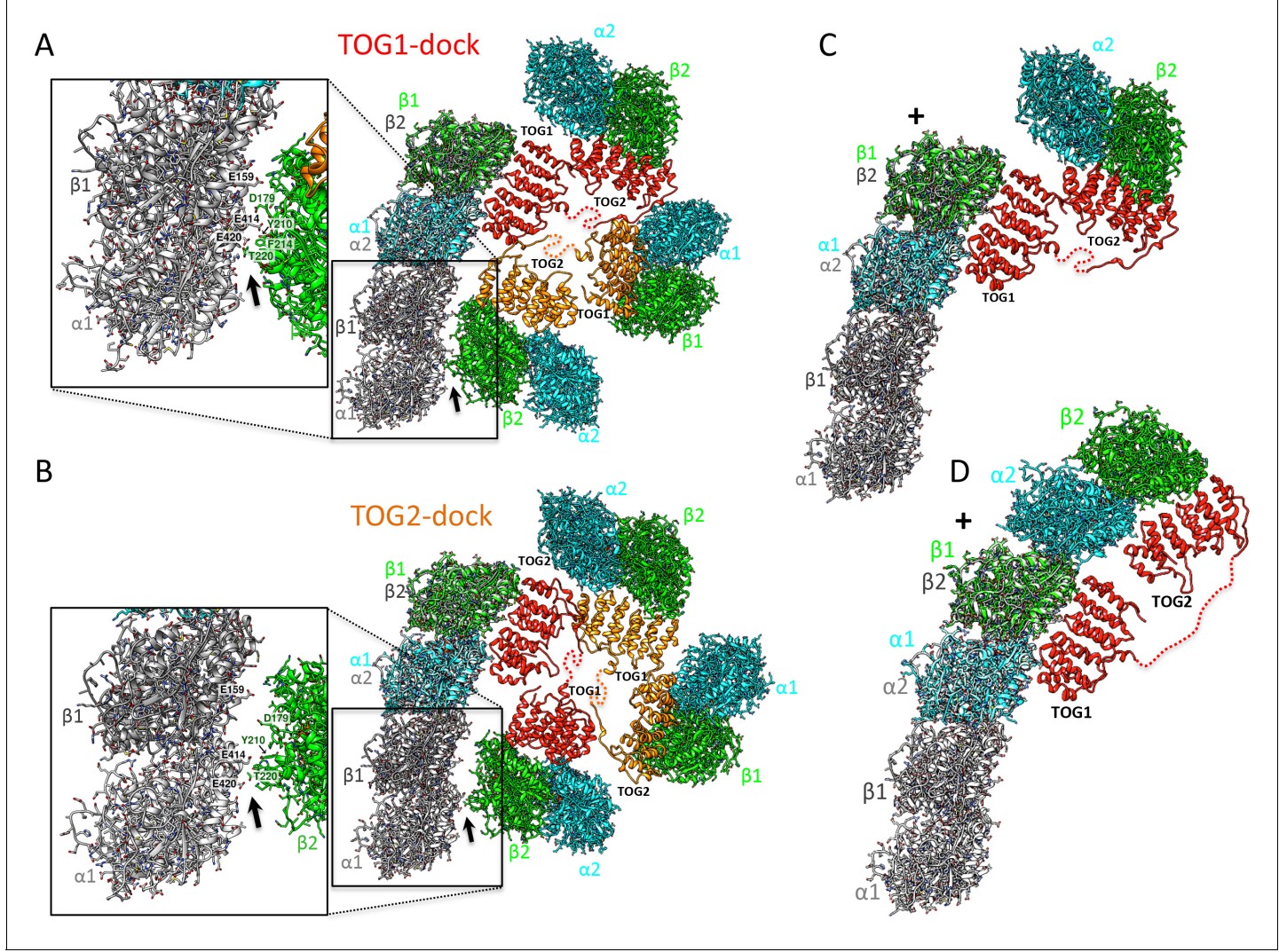

**Figure 7.** Docking of atomic structures onto protofilament ends reveals the molecular details of unfurling. (A) Right, atomic model for four αβ-tubulin-bound TOG square X-ray structures (*Figure 2*) docked using αβ-tubulin bound to TOG1 at the terminal αβ-tubulin in a curved protofilament (PDB ID: 3RYH). Left, magnified view of the zone of steric contact between TOG2-αβ-tubulin in the second subunit and the penultimate αβ-tubulin of the protofilament below the polymerization site. (B) Right, atomic model for four αβ-tubulin-bound TOG square X-ray structure (*Figure 2*) docked using αβ-tubulin bound to TOG2 at the terminal αβ-tubulin in a curved protofilament (PDB ID: 3RYH). Left, magnified view of the zone shown in A between TOG1-αβ-tubulin in the second subunit and the penultimate αβ-tubulin of the protofilament. Details and overlay images are shown in *Figure 7—figure supplement 1*. (C) Docking the isolated TOG1-TOG2 two-αβ-tubulin assembly structure (extracted from the pseudo-dimer structure) onto the terminal αβ-tubulin of the curved protofilament. (D) Docking of the unfurled 1:2:1 unfurled assembly structure (*Figure 5*) to the curved protofilament revealing TOG1 to be positioned at the base of the new assembly while TOG2 is positioned at the outer end of the newly formed MT plus-end.

DOI: https://doi.org/10.7554/eLife.38922.023

The following figure supplement is available for figure 7:

**Figure supplement 1.** All-atom docking models for structures superimposed onto curved protofilament plus-ends.

DOI: https://doi.org/10.7554/eLife.38922.024

αβ-tubulin destabilizes the TOG square in contrast with TOG2:αβ-tubulin, which will not destabilize the TOG square at protofilament tips, leading to selection of TOG1 at the docking site (Figurer 8B-II). 4) The MT plus-end-induced TOG square destabilization likely triggers TOG square disassembly into two corner-shaped TOG1-TOG2 subunits at MT plus-ends. 5) Corner-like half-square TOG1-TOG2 subunit assemblies are then released which are likely unstable (*Figure 8B–II*), and interface 2 likely acts as a flexible hinge for TOG2 to freely rotate around TOG1, driven by Brownian motion. Reversible unfurling, or hinge rotation, promotes αβ-tubulin bound to TOG2 to polymerize

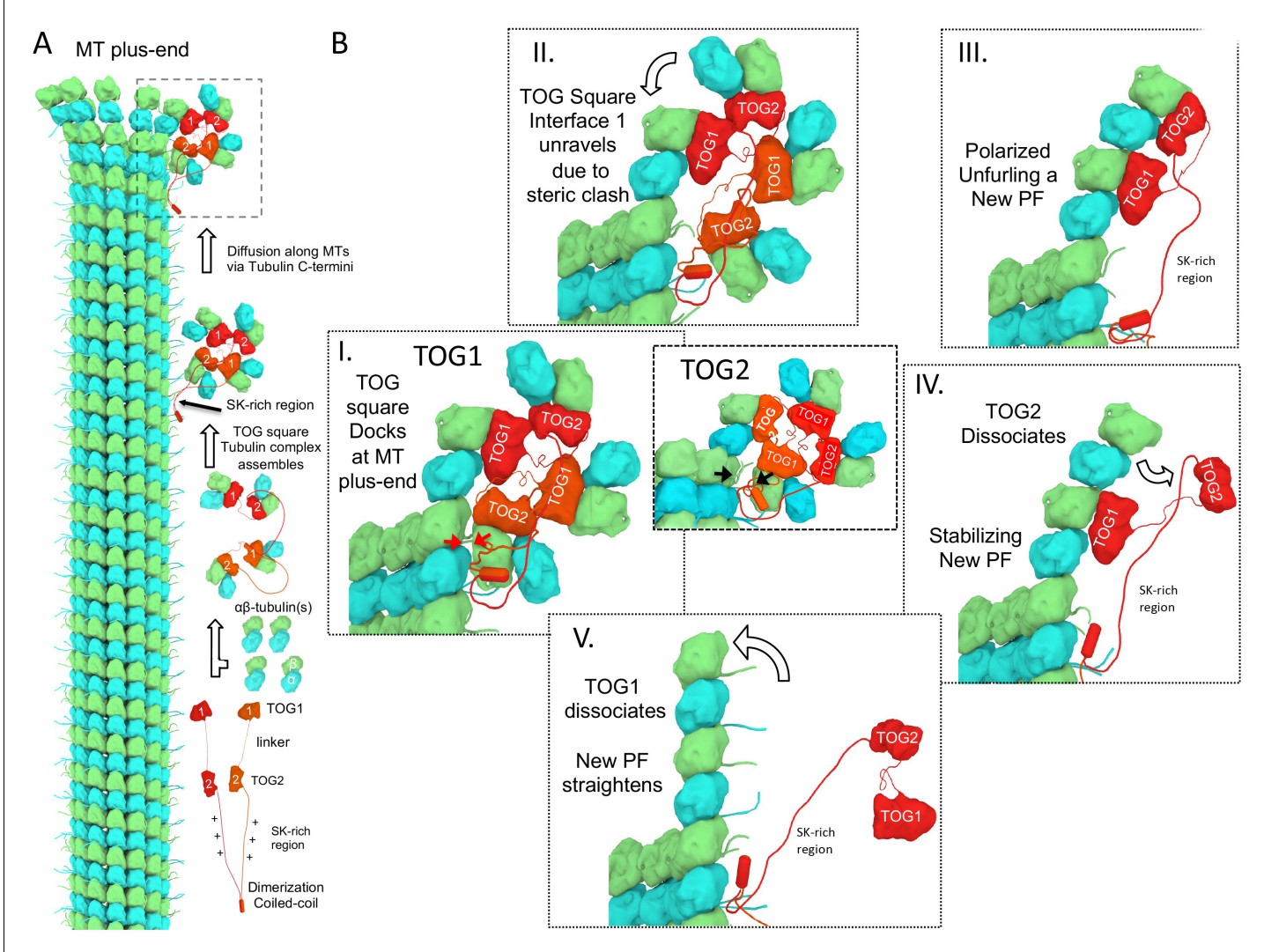

**Figure 8.** A polarized unfurling model for TOG arrays as MT polymerases. An animation for this model is shown in *Video 1*. (**A**) Assembly of yeast MT polymerase dimeric TOG1-TOG2 subunits with four αβ-tubulins into an αβ-tubulin:TOG square. TOG squares diffuse along MT lattices modulated by tubulin C-termini interacting with SK-rich regions. (**B**) I. TOG square assemblies orient αβ-tubulins in wheel-shaped assemblies at MT plus-ends. II. These assemblies are destabilized upon TOG1-α-tubulin polymerizing onto the exposed β-tubulin at MT plus-ends, releasing TOG1-TOG2 subunits in corner conformations. III. The release of TOG2:αβ-tubulin allows free rotation around TOG1, driving two αβ-tubulins to polymerize. IV. TOG2 dissociates from the newly polymerized αβ-tubulin stabilizing protofilament at the plus-end while TOG1 anchors this αβ-tubulin onto the MT plus-end. V. Straightening of this new protofilament leads to the dissociation of TOG1. The rebinding of TOG1-TOG2 subunits to αβ-tubulins reforms the TOG square assembly and restarts the MT polymerase cycle. Atomic views for states I, II, and III are shown in *Figure 7*.
DOI: https://doi.org/10.7554/eLife.38922.025

catalytically onto the plus-end of the TOG1-bound αβ-tubulin (*Figure 8B–III*). The TOG1- and TOG2-bound αβ-tubulins polymerize in a single concerted unfurling event as seen in the polymerized TOG1-TOG2:αβ-tubulin X-ray structure (*Figure 5E,F*). This event effectively 'unfurls' a single curved protofilament from two αβ-tubulins pre-oriented onto an αβ-tubulin-loaded TOG1-TOG2 corner-like intermediate. No energy expenditure is required during unfurling, as reversible Brownian motion likely drives the unfurling activity. However, formation of the polymerized assembly intermediate is captured by the αβ-tubulin-αβ-tubulin inter-dimer polymerization interfaces, which become locked by the inter-dimer interface conformational change, as seen in in the polymerized state structure (*Figure 6A,B*). The αβ-tubulin inter-dimer interfaces (1650 Å² surface area in a single interface) may compete with TOG square reformation (1930 Å² in total for a TOG square). 6) A gradient in the

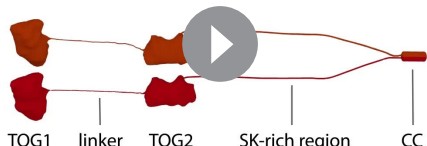

TOG1    linker    TOG2    SK-rich region    CC

**Video 1.** Animation of the mechanism of polarized unfurling for multiple TOG domains in a yeast MT polymerase. This animation describes the 'polarized unfurling' mechanism for multiple TOG domains in promoting MT polymerization. Briefly, Yeast MT polymerases are dimers with each subunit including TOG1 and TOG2 domains separated by a linker and followed by unstructured SK-rich and coiled-coil domains. Each MT polymerase binds four αβ-tubulins forming TOG square assemblies, as shown in the ribbon diagram (*Figure 2*). TOG2 exchanges αβ-tubulin due to its rapid exchange rate, while TOG square assemblies diffuse along MT lattices loaded with αβ-tubulins at MT plus-ends, as visualized in *Figure 8*. Docking of TOG square assemblies via TOG2 αβ-tubulin does not destabilize the TOG square as described in *Figure 7*. The polymerization of αβ-tubulin-TOG1 destabilizes TOG square complexes at interface 1 due to steric contact as described in *Figure 7*. The destabilization of the TOG square releases TOG2, promoting polarized unfurling of two soluble αβ-tubulins into one curved protofilament at the MT plus-end, as visualized in *Figure 5*. The newly formed protofilament is further stabilized by the rapid dissociation of TOG2 from the outermost αβ-tubulin, and forms corners to enhance direct αβ-tubulin polymerization at the MT plus-end. Other MT polymerase molecules promote delivery of αβ-tubulin dimers via the 'polarized unfurling' mechanism at polymerizing MT plus-ends. Individual steps for this model are shown in *Figure 8*.
DOI: https://doi.org/10.7554/eLife.38922.026

αβ-tubulin exchange rates between TOG1 and TOG2 likely leads TOG2 to dissociate from its αβ-tubulin rapidly before TOG1 dissociates from its αβ-tubulin (*Figure 1*; *Figure 8A–VI*). The unfurled TOG1-TOG2 αβ-tubulin polymerized structure positions this affinity gradient spatially across lower and upper positions of the polymerized complex with respect to the MT plus-end (*Figure 6*). A tightly bound αβ-tubulin on TOG1 likely anchors the TOG array onto the MT plus-end while the rapidly exchanging αβ-tubulin promotes TOG2 release- an intermediate that promotes accelerated MT polymerase (*Figure 8B* IV). 10) Protofilament straightening during MT plus-end closure likely induces TOG1 dissociation from the lower αβ-tubulin of the newly polymerized protofilament as suggested previously by the 'catch and release' model (*Ayaz et al., 2012*). The unbound TOG1-TOG2 arrays are finally released from the newly formed protofilament to reload with soluble αβ-tubulin from the cytoplasm. TOG arrays may then reform the TOG square assembly upon recruiting αβ-tubulins and restart the cycle, while the SK-rich region maintains contact with the polymerizing MT plus-end (*Figure 8B–V*) (*Video 1*).

## Implications of 'polarized unfurling'

The polarized unfurling model suggests that the MT polymerase activity of TOG arrays is due to three features: 1) pre-organization of αβ-tubulins onto the TOG square. promoted by interfaces 1 and 2. 2) MT plus-end-induced unfurling of TOG square via TOG1-αβ-tubulin induced destabilization (*Figure 7*). 3) Reversible unfurling driven by Brownian motion and the polymerization capacity of αβ-tubulins being recruited TOG1-TOG2 squares disassemble. The propensity of TOG1-TOG2 subunit head-to-tail self-assembly is strongly enhanced by C-terminal coiled-coil-dimerization or the presence of TOG3-TOG4 in metazoan TOG arrays such as XMAP215/ch-TOG proteins. The positive charge of the SK-rich region is essential to associate with the MT surface. The polarized unfurling model suggests that TOG1 and TOG2 serve specific roles in MT polymerase activity, TOG1 anchors the array and destabilizes the TOG square organization onto the MT plus-end, while TOG2 drives αβ-tubulin polymerization.

## Comparison to other MT polymerase models

Two other models were suggested for the functions of TOG arrays as MT polymerases. In these models, TOG1 and TOG2 domains were suggested to exhibit random or reversed orientations in the process of polymerizing αβ-tubulin, compared to the polarized unfurling model (*Ayaz et al., 2014*; *Fox et al., 2014*). Most recently, *Geyer et al. (2018)* studied the roles of TOG1 and TOG2 in MT polymerase activity by generating All-TOG1 or All-TOG2 chimeras of the budding yeast Stu2 MT polymerase. This study concluded that any two TOG domains, regardless of identity, are required in

a TOG array for MT polymerase activity. This observation agrees with the importance of two adjacent TOG domains being critical for forming a two αβ-tubulin polymerized complex. However, the study concluded that no higher order of organization, such as the TOG square observed here, was occurring since TOG domain identity did not influence MT polymerase activity. However, our TOG square assembly structure indicates that the critical 12-residue linker adjacent to TOG2 still remains in these All-TOG1 or All-TOG2 Stu2 constructs. Thus, our studies demonstrate that the chimeric constructs used in that study (*Geyer et al., 2018*) still retained a substantial TOG square assembly capacity, despite exchanging TOG1 for TOG2 domain sequences. These chimeric Stu2 proteins were not structurally characterized in that study, and thus their self-assembly properties using their modified TOG arrays remain unknown (*Geyer et al., 2018*).

Our cysteine mutagenesis, crosslinking/mass spectrometry, and negative stain EM studies indicate wt-Alp14-dimer forms TOG squares upon binding αβ-tubulin in solution (*Figures 3–4*). Biochemical and negative stain-EM suggest that interfaces 1 and two organize TOG square assemblies, and that their inactivation in the INT1, INT2, and INT1 +2 mutants results in specific loss of either interface 1 or interface 2 producing either single corner-like subunits bound to two orthogonally oriented αβ-tubulins, two spontaneously polymerized αβ-tubulins bound TOG1-TOG2 arrays (INT1 and INT2), or disorganized arrays composed of multiple TOG-αβ-tubulins (INT1 +2); however these defects did not influence the ability of TOG arrays to bind αβ-tubulins. Our data support that interfaces 1 and 2 pre-orient αβ-tubulins and suggest that spontaneous polymerization may occur if interface 1 or interface 2 become destabilized, supporting our model (*Figures 4–5*). The polarized unfurling model explains a well-documented observation that could not be previously rationalized. Fusion of protein masses such as GFP protein, but not short tags, onto the TOG1 N-terminus severely inactivates MT polymerases, exclusively activating their MT depolymerase activity without affecting αβ-tubulin binding (*Lechner et al., 2012*) and references herein). Our model shows that N-terminal fusions on TOG arrays strongly interfere with formation of a TOG square by forming blocks that sterically hinder interface 2 formation.

## Comparison to other MT regulatory proteins with TOG arrays

Other conserved classes of MT regulatory TOG-like array proteins, such as CLASP and Crescerin/CHE-12, may form similar TOG square-like particles and modulate MT dynamics in related mechanisms. For instance, the *S. pombe* CLASP:αβ-tubulin complexes form wheel-like particles with similar dimensions and promote local MT rescue (*Al-Bassam and Chang, 2011*; *Al-Bassam et al., 2010*; *Das et al., 2015*). However, the high-resolution organization of these different TOG arrays remains to be determined at high resolution.

# Materials and methods

**Key resources table**

| Reagent type (species) or resource | Designation | Source | Identifier | Additional information |
|---|---|---|---|---|
| Chemical compound, drug | Darpin D1 (Synthetic DNA) | Invitrogen | N/A | |
| Chemical compound, drug | GTP | Sigma | G-8877 | |
| Chemical compound, drug | GDP | Sigma | G7127 | |
| Chemical compound, drug | Crystallization plates | TTP Labtech | 4150–05600 | |
| Chemical compound, drug | Crystallization sparse matrix screens | Qiagen | N/A | |

*Continued on next page*

*Continued*

| Reagent type (species) or resource | Designation | Source | Identifier | Additional information |
|---|---|---|---|---|
| Chemical compound, drug | PEG-8000 | Sigma | 1546605 | |
| Chemical compound, drug | PEG-2000 | Sigma | 8.21037 | |
| Chemical compound, drug | Copper(II) sulfate | Sigma | C1297 | |
| Chemical compound, drug | 1, 10-phenanthroline | Sigma | 131377 | |
| Chemical compound, drug | Trypsin | Sigma | T6567 | |
| Chemical compound, drug | Chymotrypsin | Sigma | C6423 | |
| Chemical compound, drug | Iodoacetamide | Sigma | I6125 | |
| Chemical compound, drug | 4-Vinylpyridine | Sigma | V3204 | |
| Other | 2:4:4 sk-Alp14-550:αβ-Tubulin:DRP | Protein Data Bank | PDB: #6MZF | Deposited Data (Atomic coordinates) |
| Other | 2:4:4 sk-Alp14-550-SL:αβ-Tubulin: DRP | Protein Data Bank | PDB: #6MZE | Deposited Data (Atomic coordinates) |
| Other | 1:2:1 sk-Alp14-550:αβ-Tubulin:DRPΔN | Protein Data Bank | PDB: #6MZG | Deposited Data (Atomic coordinates) |
| Other | *Saccharomyces cerevisiae* Stu2p | UniprotKB/Swiss-Prot | P46675 | Protein sequence |
| Other | *Saccharomyces kluyveri* Stu2p or Alp14p | Lachancea kluyveri NRRL Y-12651 chromosome | SKLU-Cont10078 | Protein sequence |
| Other | *Schizos accharomyces pombe* Alp14p | UniprotKB/Swiss-Prot | O94534 | Protein sequence |
| Other | *Chaetomium thermophilum* Stu2 | UniprotKB/Swiss-Prot | G0S3A7 | Protein sequence |
| Other | SoluBL21 bacterial expression system | AmsBio | C700200 | Model system (expression system) |
| Recombinant DNA reagent | pLIC_V2-*Sc* Stu2p-H$_6$ | Current study | N/A | Recombinant DNA constructs Expressed in bacterial strains |
| Recombinant DNA reagent | pLIC_V2-*Sk* Stu2p-H$_6$ | Current study | N/A | |

*Continued on next page*

*Continued*

| Reagent type (species) or resource | Designation | Source | Identifier | Additional information |
|---|---|---|---|---|
| Recombinant DNA reagent | pLIC_V2-Sc Stu2-550-H$_6$ (TOG1-TOG2 monomer) | *Al-Bassam et al., 2006* | | |
| Recombinant DNA reagent | pLIC_V2-KL-Stu2 -monomer-H$_6$ (residues 1–560) | Current study | NA | |
| Recombinant DNA reagent | pLIC_V2-CT Stu2-mon omer-H$_6$ (residues 1–550) | Current study | NA | |
| Recombinant DNA reagent | pLIC_V2-SK Alp14-monomer-H$_6$ (residues 1-550) | Current study | N/A | |
| Recombinant DNA reagent | pLIC_V2-Sk Alp14-monomer -SL-H$_6$ (residues 1–550; linker residues replaced KL sequence; see Materials and methods) | Current study | N/A | |
| Recombinant DNA reagent | pLIC_V2-sk-wt-Alp14-dimer-H6 (residues 1–724) | Current study | N/A | |
| Recombinant DNA reagent | pLIC_V2 -sk-Alp14-dimer- H$_6$ | Current study | N/A | |
| Recombinant DNA reagent | S180C and L304C (residues 1–724) | | | |
| Recombinant DNA reagent | pLIC_V2-sk-Alp14-dimer- H$_6$ | Current study | N/A | |
| Recombinant DNA reagent | S41C and E518C (residues 1–724) | | | |
| Recombinant DNA reagent | pLIC_V2-wt -Alp14-dimer-H$_6$ (residues 1–690) | Current study | N/A | |
| Recombinant DNA reagent | pLIC_V2-TOG1M - H$_6$ (residues 1–690: Y23A and R23A) | Current study | N/A | |
| Recombinant DNA reagent | pLIC_V2-TOG2M - H$_6$ (residues 1–690: Y300A and K381A) | Current study | N/A | |

*Continued on next page*

Continued

| Reagent type (species) or resource | Designation | Source | Identifier | Additional information |
|---|---|---|---|---|
| Recombinant DNA reagent | pLIC_V2-INT1-$H_6$ (residues 1–690: L206A, L208A, F275R D276A, L277A, V278A, K320L, R359A) | Current study | N/A | |
| Recombinant DNA reagent | pLIC_V2-INT2-$H_6$ (residues 1–690: L39D, S40A, D42A, L437D, S440A, E478A and R479A) | Current study | N/A | |
| Recombinant DNA reagent | pLIC_V2 -INT1 + 2 $H_6$ (L206A, L208A, F275R D276A, L277A, V278A, K320L, R359A L39D, S40A, D42A, L437D, S440A, E478A and R479A) | Current study | N/A | |
| Recombinant DNA reagent | pET303-$H_6$-DRP | Current study | N/A | |
| Recombinant DNA reagent | pLIC_V2-$H_6$-DRPΔN | Current study | N/A | |
| Other | αβ-tubulin purified from porcine brains | Current study | N/A | Native protein purification |
| Other | αβ-tubulin purified from porcine brains | *Castoldi and Popov, 2003* | | |
| Software, algorithm | ASTRA V6.0 | Wyatt Technology | http://www.wyatt. com/products/ software/astra.html | |
| Software, algorithm | NanoAnalyze | TA Instruments | http://www.t ainstruments.com/ | |
| Software, algorithm | EMAN2 | | http://blake.bcm .edu/emanwiki/EMAN2 | |
| Software, algorithm | iMOSFLM | *Battye et al., 2011* | http://www. mrc-lmb.cam.ac. uk/harry/imosflm/ver7 21/quickguide.html | |
| Software, algorithm | PHASER | *Terwilliger, 2000* | http://www.p haser.cimr.cam.ac. uk/index.php/ | |
| Software, algorithm | PHASER | *McCoy, 2007* | Phaser _Crystallographic_ Software | |
| Software, algorithm | PyMol | Schrodinger, LLC | http://www .pymol.org/ | |
| Software, algorithm | UCSF-Chimera | *Pettersen et al., 2004* | https://www.c gl.ucsf.edu/chimera/ | |
| Software, algorithm | DM from CCP4 suite | *Cowtan and Main, 1996* | http://ww w.ccp4.ac.uk/html/ dmmulti.html | |

*Continued on next page*

*Continued*

| Reagent type (species) or resource | Designation | Source | Identifier | Additional information |
|---|---|---|---|---|
| Software, algorithm | PHENIX | *Adams et al., 2010* | https://www.phenix-online.org | |
| Software, algorithm | anisotropy server | *Strong et al., 2006* | https://services.mbi.ucla.edu/anisoscale/ | |
| Software, algorithm | Phyre protein homology model | *Kelley et al., 2015* | www.sbg.bio.ic.ac.uk/phyre2/html/page.cgi?id=index | |
| Software, algorithm | Cr-yolo | *Wagner et al., 2018* | http://sphire.mpg.de/wiki/ | |
| Software, algorithm | Relion 2.2 | *Kimanius et al., 2016* | https://www2.mrc-lmb.cam.ac.uk/relion/index.php | |
| Software, algorithm | Cryosparc | *Punjani et al., 2017* | https://cryosparc.com/ | |
| Software, algorithm | MolProbity | *Chen et al., 2012* | http://molprobity.biochem.duke.edu | |
| Software, algorithm | Coot | *Emsley et al., 2010* | http://www2.mrc-lmb.cam.ac.uk/personal/pemsley/coot/ | |
| Software, algorithm | BLENDER 3D-animation | Blender foundation | https://www.blender.org/ | |

## Protein expression and purification of Alp14 and sk-Alp14 proteins

The coding regions for MT polymerases from *S. pombe* Alp14p (accession: BAA84527.1), *S. cerevisiae* Stu2p (accession: CAA97574.1), *Saccharomyces kluyveri* Alp14 or Stu2p (coding region identified in accession: SKLU-Cont10078), and *Chaetomium thermophilum* Stu2p (accession: XP_006692509) were inserted into bacterial expression vectors with a C-terminal His-tag. wt-Alp14-monomer (residues 1–510), wt-Alp14-dimer (residues 1–690), sk-Alp14-monomer (residues 1–550), sk-Alp14-dimer (residues 1–724), Sc-Stu2-dimer (residues 1–746), and Ct-Stu2-dimer (residues 1–719) constructs were generated, including with or without the SK-rich and coiled-coil dimerization regions. TOG1M and TOG2M mutants were generated via point mutagenesis of Y23A and R23A to inactivate TOG1 domains (TOG1M); and Y300A and K381A to inactivate TOG2 domains (TOG2M) (*Al-Bassam et al., 2012*). The INT1 +2 mutant was generated via gene synthesis (Epoch Life Science) by introducing 15-residue mutations into the wt-Alp14-dimer sequence (L206A L208A F275R D276A L277A V278A K320L R359A L39D S40A D42A L437D S440A E478A R479A). The INT1 and INT2 mutants were generated by a PCR swapping strategy of INT1 +2 with wt-Alp14-dimer leading to INT1 with 8-residue mutations (L206A L208A F275R D276A L277A V278A K320L R359A) and INT2 with 7-residue mutations (L39D S40A D42A L437D S440A E478A R479A). Generally, constructs were transformed and expressed in BL21 bacterial strains using the T7 expression system, and were grown at 37°C and induced with 0.5 mM isopropyl thio-β-glycoside at 18°C overnight. Cells were centrifuged and then lysed using a microfluidizer (Avastin). Extracts were clarified via centrifugation at 18,000 x *g*. Proteins were purified using Ni-IDA (Macherey-Nagel) and/or ion exchange using Hitrap-SP or Hitrap-Q chromatography followed by size exclusion chromatography using a Superdex 200 (30/1000) column (GE Healthcare). DRP was synthesized (Gene Art, Life Technologies), inserted into bacterial expression vectors with a C-terminal 6 × His tag, and expressed as described above. Proteins were purified using Ni-NTA (Macherey-Nagel) followed by Hitrap Q ion exchange and followed by size exclusion chromatography as described above. Purified proteins were used immediately or frozen in liquid nitrogen for future use.

## Biochemical analyses of Alp14:αβ-tubulin complexes

Soluble porcine αβ-tubulin (10 μM or 20 μM) purified using two GTP-polymerization cycles at high ionic strength as previously described (*Castoldi and Popov, 2003*) was mixed with 5 μM *S. kluyveri* (sk) or *S. pombe* wt-Alp14-monomer, wt-Alp14-dimer, TOG1M, TOG2M, INT1, INT2, or INT1 +2 mutant proteins and then diluted five-fold. To assess αβ-tubulin assembly, the protein mixtures were analyzed by mixing the proteins into 0.5 mL volumes and injecting them into a Superdex 200 (10/300) size exclusion chromatography (SEC) column equilibrated in 100 mM or 200 mM KCl binding buffer (50 mM HEPES [pH 7.0], 1 mM MgCl$_2$, and 1 mM β-mercaptoethanol with 100 mM KCl or 200 mM KCl) using an AKTA purifier system (GE Healthcare). Elution fractions (0.5 mL) were collected and analyzed via sodium dodecyl sulfate polyacrylamide gel electrophoresis (Bio-Rad). The αβ-tubulin- and Alp14-containing bands were quantitated using densitometry to determine the amounts of bound and unbound αβ-tubulin in each SEC fraction. Molecular masses of wt-Alp14-monomer, wt-Alp14-dimer, TOG1M, TOG2M, INT1, INT2, and INT1 +2 proteins, αβ-tubulin, and their complexes were measured using SEC-coupled multi-angle light scattering (SEC-MALS). Complexes were separated on Superdex 200 Increase (10/300) columns (GE Healthcare) while measuring UV absorbance (Agilent 1100-Series HPLC), light scattering (Wyatt Technology miniDAWN TREOS), and refractive index (Wyatt Technology Optilab T-rEX). Concentration-weighted molecular masses for each peak were calculated using ASTRA six software (Wyatt Technology).

Isothermal titration calorimetry (ITC) was performed using a Nano-ITC (TA Instruments) to determine DRP and DRPΔN affinities for αβ-tubulin. Experiments were performed at 25°C. Soluble αβ-tubulin, DRP, and DRPΔN were diluted in 50 mM HEPES buffer, pH 7.3, 100 mM KCl, 1 mM MgCl$_2$, and 50 μM GDP. The sample cell was filled with tubulin (20–40 μM) for every experiment. 135–250 μM of DRP or DRPΔN solutions were injected in volumes of 2 or 5 μL in a series of controlled doses into the sample cell. To determine TOG1 and TOG2 affinities for αβ-tubulin with DRP, proteins were diluted in 50 mM HEPES buffer, pH 7.3, 100 or 200 mM KCl, and 1 mM MgCl$_2$. 100–250 μM of TOG1 or TOG2 solutions were injected in volumes of 2 or 5 μL in a series of controlled does into the sample cell containing 1:1 molar ratio of αβ-tubulin and DRP (20–40 μM). The results were analyzed with NanoAnalyze software (TA Instruments). Thermodynamic parameters were calculated using the standard thermodynamic equation: $-RT\ln K_a = \Delta G = \Delta H - T\Delta S$, where $\Delta G$, $\Delta H$, and $\Delta S$ are the changes in free energy, enthalpy, and entropy of binding, respectively, $T$ is the absolute temperature, and $R$ is the gas constant (1.98 cal mol$^{-1}$ K$^{-1}$).

## Crystallization of sk-Alp14:αβ-tubulin:DRP or drpδn complexes

Complexes (200 μM) were screened for crystallization using commercial sparse matrix (Qiagen) or homemade screens in 96-well format using a Mosquito robot (TTP Labtech) via the hanging drop method. Cube-shaped crystals (5 μm on each edge) formed for sk-Alp14-monomer:αβ-tubulin:DRP complexes and grew over 4–7 days in 50 mM PIPES, 100 mM MgCl$_2$ [pH 7.0], and 10–15% PEG-8000. Larger crystals were grown using micro-seeding (*Figure 2—figure supplement 2A*). To obtain improved X-ray diffraction (see below), we used an sk-Alp14-monomer construct in which non-conserved 256–297 residue linkers were replaced by the shorter linker (including the residue sequence -AVPAQSDNNSTLQTDKDGDTLMGN-) from the *K. lactis* ortholog sequence (termed sk-Alp14-monomer-SL). Crystals were transferred to 50 mM PIPES, 100 mM MgCl$_2$ [pH 7.0], 15% PEG-8000, and 25% glycerol for cryo-protection and flash frozen in liquid nitrogen.

Rectangular crystals of sk-Alp14-monomer:αβ-tubulin:DRPΔN complexes formed in 7–10 days under the same conditions described for cube-shaped TOG1-TOG2:αβ-tubulin:DRP crystals. These rectangular crystals exclusively formed using DRPΔN (did not form with DRP) and were obtained using a variety of constructs of monomeric as well as dimeric sk-Alp14-monomer (*Table 3*). Rectangular sk-Alp14-monomer:αβ-tubulin:DRPΔN crystals were treated for cryo-protection and flash frozen as described above.

## X-ray diffraction and structure determination of sk-Alp14:αβ-tubulin assemblies

More than 100 sk-Alp14-monomer:αβ-tubulin:DRP crystals were screened for X-ray diffraction at the Argonne National Laboratory at the Advanced Photon Source microfocus 24-ID-C beamline. Anisotropic X-ray diffraction data were collected for the best cube-shaped crystals in the *P*2$_1$ space group

to 4.4 Å resolution in the best dimension, with unit cell dimensions $a$ = 219 Å, $b$ = 108 Å, and $c$ = 283 Å (*Figure 2—figure supplement 1*). The sk-Alp14-monomer-SL:αβ-tubulin:DRP crystals showed improved diffraction and decreased anisotropy to 3.6 Å resolution in an identical $P2_1$ unit cell (*Table 3*). X-ray diffraction data were indexed and scaled using iMOSFLM and treated for aniso-tropic diffraction using ellipsoidal truncation on the UCLA diffraction anisotropy server (services.mbi.ucla.edu/anisoscale). Phase information was determined using TOG1 (PDB ID:4FFB), TOG2 (PDB ID:4U3J), αβ-tubulin dimer, and DRP (PDB ID:4DRX) models using molecular replacement. Briefly, a truncated poly-alanine TOG domain including only its HEAT repeats was used in the molecular replacement rotation and translation search (*Figure 2—figure supplement 1B–C*). Eight αβ-tubulin and TOG domain solutions were identified based on the Log Likelihood Gain (LLG) values (*Figure 2—figure supplement 1B–C*). After eight cycles of density modification, the electron-density map revealed the TOG1 domains exclusively due to the unique C-terminal linker and vertical helix densities (*Figure 2—figure supplement 1D–E*). Density for eight DRP molecules was identified and built. DRP molecules interacted only with their cognate β-tubulin and did not form interfaces with α-tubulin from neighboring molecules (*Figure 2—figure supplement 1H*). Two 2:4:4 sk-Alp14-mono-mer:αβ-tubulin:DRP wheel-like models were built and subjected to cycles of rigid-body refinement and model building using the *S. kluyveri* ortholog sequence. Each asymmetric unit contained two wheel-like assemblies (*Figure 2—figure supplement 1E*). TOG1-TOG2 linker residues (residues 265–299 in native sk-Alp14 and residues 260–277 in sk-Alp14-SL) were not observed and were pre-sumed to be disordered (*Figure 2—figure supplement 1F–H*). Density maps from each of the wheel-like models were averaged using non-crystallographic symmetry and then refined using the PHENIX program (*Adams et al., 2010*). Initially, models were refined using non-crystallographic sym-metry (16 fold NCS) restraints and strictly constrained coordinates with group B-factor schemes. In the final stage refinement, the strategy was switched to individual positional and isotropic B-factor with automatic weight optimization. A 4.4 Å sk-Alp14-monomer:αβ-tubulin:DRP structure and 3.6 Å sk-Alp14-monomer-SL:αβ-tubulin:DRP structure are reported; refinement statistics appear in *Table 3*.

Rectangular crystals formed from sk-Alp14-monomer:αβ-tubulin: DRPΔN diffracted to 3.2 Å reso-lution at the Argonne National Laboratory at the Advanced Photon Source microfocus APS 24-ID-C beamline. X-ray diffraction data were indexed in the $P2_1$ space group with unique unit cell dimen-sions $a$ = 115 Å, $b$ = 194 Å, and $c$ = 149 Å, with two complexes in each unit cell (*Table 3*). Phase information was determined using molecular replacement using the TOG1 and TOG2 domains and curved αβ-tubulin as search models (*Figure 5—figure supplement 1B*). TOG1 and TOG2 domains were identified after cycles of density modification as described above. Four αβ-tubulins, four TOG domains, and two DRPΔN models were placed in the unit cell. The identity of TOG domains was determined using the conserved C-terminal linker and jutting helix in the TOG1 domain sequence. A single DRPΔN molecule was identified bound per two αβ-tubulin polymerized complex. Data from each extended assembly were combined using non-crystallographic symmetry (8-fold NCS) and were averaged and refined using the program PHENIX (*Adams et al., 2010*) (*Table 3*). The individ-ual positional coordinates and anisotropic B-factor were refined with automatic weight optimization in the final stage. A 3.2 Å resolution refined density map is presented in *Figure 5—figure supple-ment 1C*. Examining data quality of sk-Alp14-monomer:αβ-tubulin:DRP or sk-Alp14-monomer:αβ-tubulin:DRPΔN using PHENIX (*Adams et al., 2010*) indicated that the diffraction data contained a small degree of pseudo-merohedral twinning. The twin fractions were adjusted during refinement of both models.

## Cysteine mutagenesis and crosslinking analyses of sk-Alp14:αβ-tubulin assemblies

Based on the sk-Alp14-monomer:αβ-tubulin:DRP crystal structure, the *S. kluyveri* ortholog protein sk-Alp14, in its dimer form (residues 1–724), was used to generate crosslinking mutations. Interface 1 residues, which are in close proximity to each other, were mutated to cysteine: Ser180Cys (S180C) and Leu304Cys (L304C), which we termed S180C-L304C. Interface 2 residues, which are in close proximity to each other, were also mutated to cysteine: Ser41Cys (S41C) and Glu518Cys (E518C), which we termed S41C-E518C. The *S. kluyveri* ortholog dimer S180C-L304C mutant and S41C-E518C mutant proteins were purified as described above (*Figure 3E*). These constructs were used either directly or to make complexes with αβ-tubulin in a 2:4 (subunit:αβ-tubulin) molar ratio, as

described in *Figure 1A*. These S180C-L304C and S41C E518C mutants or their αβ-tubulin complexes were then treated using 5 mM Cu-phenanthroline in 50 mM HEPES and 100 mM KCl, pH 7.0, for 5 min, then treated with 5 mM EDTA. These protein mixtures were subjected to SDS-PAGE under oxidizing conditions.

For LC/MS-MS mass spectrometry-based disulfide peptide mapping, S180C-L304C sk-Alp14 oxidized SDS-PAGE bands were subjected to in-gel proteolysis using either trypsin or chymotrypsin. Fragmented peptides were then purified and treated with 5 mM iodioacetamide, which covalently adds 57 Da in mass onto reduced cysteine-containing peptides, and does not affect cysteines locked in disulfides. The peptide mixture was then treated with 5 mM dithiothriatol to reduce disulfides and then treated with 5-vinyl chloride, which covalently adds 105 Da mass units onto newly reduced cysteine-containing peptides. LCMS/MS mass spectrometry was performed and the resulting peptides were analyzed. Peptides covering 90% of sk-Alp14 were identified as were the majority of cysteines. Only two peptides were identified with cysteine residues that included 105 Da mass units added as described in *Figure 3—figure supplement 1B*.

## Negative stain electron microscopy and image analysis of wt-Alp14 and TOG inactivated mutants in complex with αβ-tubulins

SEC-purified αβ-tubulin complexes of wt-Alp14-dimer, INT1, INT2, and INT1 +2 at 100 mM KCl supplemented with glutaraldehyde 0.05% and 4:2 molar ratio were placed on glow discharged grids, blotted after 30–60 s, and then stained with multiple washes of 0.1% uranyl formate at pH 7.0. All the images for negatively stained specimens were grids were collected on an electron microscope (JEM-2100F; JEOL) equipped with a field emission gun using low-dose mode at 200 KeV paired with a DE-20 direct electron detector device (DDD) operating in integration mode. The images were then processed using neural networks picking using Cr-YOLO (Wagner et al, https://doi.org/10.1101/356584). The particle coordinates were imported into relion 2.2. Images were CTF corrected using CTFFIND4 (Grant et al,). Particles were manually screened and subjected to rounds of 2D-classification either in Relion 2.2 or using Cryosparc (*Punjani et al., 2017*; *Kimanius et al., 2016*). For each data set, 2D-Class averages were grouped based on their conformation then compared to 30 Å resolution filtered models of 4:2 αβ-tubulin: TOG2-TOG2 in square conformation (*Figure 2*) and single TOG1-TOG2 subunit from the square conformation in the bent conformation bound to two non-polymerized αβ-tubulins (*Figure 6*), TOG1-TOG2 in the polymerized conformation (*Figure 5*) and a single TOG domain bound αβ-tubulin (PDB-ID:4FFD). The projection matching was performed using EMAN2 command e2classvsproj.py at 1–5 angular degree increments (*Tang et al., 2007*).

## Animating the MT polymerase 'polarized unfurling' mechanism

The animation was created using BLENDER 3D-animation software (http://blender.org) as follows. Briefly, surface and ribbon models of PDB coordinates representing the structures were exported from UCSF-Chimera and imported into BLENDER, and then smoothed and optimized to generate animated models. Additional protein SK-rich regions and coiled-coil domains, whose structures are unknown, were thus modeled using sequence length and other information as guidance. The microtubule lattice was modeled based on the tubulin structure (PDB ID 3J6F). The dissociation of TOG1 and TOG2 domains from αβ-tubulins were simulated in the animation, based on biochemical studies described in *Figure 1* and its figure supplements.

## Acknowledgements

We thank Dr. Julian Eskin (Brandeis University) for animating the microtubule polymerase mechanism. We thank Advanced Photon Source (APS) and Drs. K Rajashankar, J Schuermann, N Sukumar, and D Neau of the Northeastern Collaborative Access Team (NE-CAT) for use of the 24-ID-C and ID-E beam lines to collect all X-ray diffraction data for our crystallographic studies. We thank Dr. Christopher Fraser (UC Davis) for using his Nano-ITC. We thank Dr. Rick Mckenney (UC Davis), and Dr. Kevin Corbett (UC San Diego) for advice and critical reading of this manuscript. JAB and FC are supported by National Institutes of Health GM110283 and GM115185, respectively. This work is based upon research conducted at the NE-CAT beamlines, which are funded by the National Institute of General Medical Sciences from the National Institutes of Health (P41 GM103403). The Pilatus 6M detector on 24-ID-C beam line is funded by a NIH-ORIP HEI grant (S10 RR029205). The Eiger

16M detector on 24-ID-E beam line is funded by a NIH-ORIP HEI grant (S10OD021527). This research used resources of the Advanced Photon Source, a US Department of Energy (DOE) Office of Science User Facility operated for the DOE Office of Science by Argonne National Laboratory under Contract No. DE-AC02-06CH11357. sk-Alp14-monomer:αβ-tubulin:DRP, sk-Alp14-monomer-SL:αβ-tubulin:DRP, and sk-Alp14-monomer:αβ-tubulin:DRPΔN are available at the protein data bank (PDB) under PDB-ID 6MZF, PDB-ID 6MZE and PDB-ID 6MZG, respectively.

## Additional information

### Funding

| Funder | Grant reference number | Author |
| --- | --- | --- |
| National Science Foundation | MCB1615991 | Jawdat Al-Bassam |
| National Institutes of Health | GM115185 | Jawdat Al-Bassam |
| National Institutes of Health | GM110283 | Jawdat Al-Bassam |

The funders had no role in study design, data collection and interpretation, or the decision to submit the work for publication.

### Author contributions

Stanley Nithianantham, Conceptualization, Data curation, Software, Formal analysis, Supervision, Validation, Investigation, Visualization, Methodology, Writing—original draft, Writing—review and editing; Brian D Cook, Conceptualization, Resources, Data curation, Software, Formal analysis, Supervision, Funding acquisition, Validation, Investigation, Visualization, Methodology, Writing—review and editing; Madeleine Beans, Fei Guo, Data curation; Fred Chang, Conceptualization, Supervision, Funding acquisition, Validation, Methodology, Writing—original draft, Project administration, Writing—review and editing; Jawdat Al-Bassam, Conceptualization, Resources, Data curation, Software, Formal analysis, Supervision, Funding acquisition, Validation, Investigation, Visualization, Methodology, Writing—original draft, Project administration, Writing—review and editing

### Author ORCIDs

Stanley Nithianantham http://orcid.org/0000-0001-6238-647X
Jawdat Al-Bassam http://orcid.org/0000-0001-6625-2102

### Decision letter and Author response

Decision letter https://doi.org/10.7554/eLife.38922.035
Author response https://doi.org/10.7554/eLife.38922.036

## Additional files

### Supplementary files

- Transparent reporting form
DOI: https://doi.org/10.7554/eLife.38922.027

### Data availability

All data generated or analyzed during this study are included in the manuscript and supporting files.

The following datasets were generated:

| Author(s) | Year | Dataset title | Dataset URL | Database and Identifier |
| --- | --- | --- | --- | --- |
| Stanley Nithianantham, Brian D Cook, Madielene Beans, Fei Guo, Fred Chang, Jawdat Al- | 2018 | 1:2:2 sk-Alp14-momomer: $\alpha\beta$-Tubulin:DRP | https://www.rcsb.org/structure/6MZF | RCSB Protein Data Bank, 6MZF |

| Bassam | | | | |
|---|---|---|---|---|
| Stanley Nithianantham, Brian D Cook, Madielene Beans, Fei Guo, Fred Chang, Jawdat Al-Bassam | 2018 | 1:2:2 sk-Alp14-monomer-SL: $\alpha\beta$-Tubulin:DRP | https://www.rcsb.org/structure/6MZE | RCSB Protein Data Bank, 6MZE |
| Stanley Nithianantham, Brian D Cook, Madielene Beans, Fei Guo, Fred Chang, Jawdat Al-Bassam | 2018 | 1:2:1 sk-Alp14-monomer: $\alpha\beta$-Tubulin:DRP$\Delta$N | https://www.rcsb.org/structure/6MZG | RCSB Protein Data Bank, 6MZG |

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
