## [Decision Letter]

Thank you for submitting your article "Structural Basis of Tubulin Recruitment and Assembly by Tumor Overexpressed Gene domain array Microtubule Polymerases" for consideration by *eLife*. Your article has been reviewed by Andrea Musacchio as the Senior Editor a Reviewing Editor, and three reviewers. The reviewers have opted to remain anonymous.

The same three reviewers evaluated also the accompanying manuscript, as you have suggested.

The reviewers have discussed the reviews with one another and the Reviewing Editor has drafted this decision to help you prepare a revised submission.

Summary:

In this first manuscript, crystal structures of two TOG domains from Alp14, bound to tubulin dimers, are presented to shed new light on how TOG domains stimulate microtubule polymerization. In one structure tubulin polymerization is blocked by the presence of a Darpin with high affinity for tubulin. This structure reveals a new and interesting square arrangement of 4 TOG domains (as found in an Alp14 dimer). The authors show by cross linking that this arrangement exists in solution. Using a Darpin with lower affinity the authors determine a structure in which the TOG domains have rearranged so that tubulin dimers are brought together ready for polymerization. Together with biochemical studies, which show a difference in affinity between TOG1 and TOG2 of Alp14, the authors put together a new model for how TOG domains catalyse microtubule growth.

The manuscript contains much that is new and interesting and it is a good candidate for *eLife*. The reviewers had however some major concerns that should be addressed.

Essential revisions:

1) The negative stain EM data (Figure 4F-I) needs to be improved. The blobs shown in Figure 4F are similar to those observed with badly behaved samples in EM and there is currently no evidence they correspond to the Alp14 structure. The authors would need to be able to show 2D averages of the WT sample in order to use negative stain analysis and compare the averages to projections of the crystal structure. The particles in images in H and I are too close together. There is no evidence that the particles in the white boxes correspond to one Alp14 complex. The data would need to be collected at lower dilutions so individual particles are clear. It would also be important to provide 2D classification on these images.

2) The evidence for the square form of the Alp14/tubulin complex playing a role in microtubule polymerization should be presented in this manuscript (particularly given that the reviewers did not support publication of the accompanying manuscript in *eLife* – see the Decision regarding the accompanying manuscript). The authors should explain why a rather large number of mutations were introduced to disrupt the interfaces. The reviewers expected that a smaller number of mutations could have been sufficient and wondered whether effects other than disrupting the interfaces could lead to the defects in microtubule polymerisation.

3) In parts the amount of the presented data are rather overwhelming particularly given that not all of the data are described. The presentation of these data needs to be improved. The data shown in Figure 1 and the related supplementary figures are only partly described. Therefore, this part of the manuscript requires hours of work to really understand which data support which claims. A better way to present these data could be to present in the main Figure 1: a graph summarising the most important MALS data for some complexes without Darpin (which ultimately are the strongest evidence for binding stoichiometries), Figure 1E and G; and present everything else in supplementary figures including a clear description/summary in their legends of what one sees in these figures.

4) The presentation of the model in the Discussion section needs improvement. The current description of the model in Figure 8 gives the impression that the model is very speculative, probably because the description of the model is combined with the implications. The authors are encouraged to first describe the model in its entirety, then reiterate the supporting evidence for it, and only then discuss the implications.

5) A recent publication (Geyer et al., 2018) provides data and a model that do not appear to be in agreement with the model proposed here. The authors should discuss these recently published findings and adjust claims as appropriate in the light of these new data.

[Editors' note: further revisions were requested prior to acceptance, as described below.]

Thank you for resubmitting your work entitled "Structural Basis of Tubulin Recruitment and Assembly by Microtubule Polymerases with Tumor Overexpressed Gene (TOG) Domain Arrays" for further consideration at *eLife*. Your revised article has been favorably evaluated by Andrea Musacchio (Senior Editor), a Reviewing Editor, and the two original reviewers.

The manuscript has been improved. Particularly the improved negative stain EM part has strengthened the manuscript and has satisfactorily addressed a major concern of the reviewers. All reviewers also welcome the other changes, particularly the editorial changes that have improved clarity of presentation in most parts. The reviewers accept the decision of the authors not to combine the biochemical data demonstrating functional importance of the interfaces involved in 'square assembly' formation, with the structural data presented in this manuscript, although it would have strengthened the arguments made.

But there are some remaining issues that need to be addressed before acceptance.

Despite parts of the Discussion being clearer now, all reviewers agree that the Discussion can be considerably more concise. The discussion of the Cook et al. paper can be deleted, and the discussion of the Geyer et al., paper should be more balanced. The statement "all-TOG1 or TOG2 chimeric constructs retained an MT polymerase activity level that was substantially lower than native Stu2" seems to be an oversimplification, since Geyer et al. reported a 84% activity level for the TOG2-TOG2 construct, which is remarkably close to the wildtype level.

We therefore invite you submit a revised version of your manuscript with a more balanced and importantly substantially shorter discussion, focusing on the main points relevant to this study. We also ask to consider addressing previously stated minor concerns that have not been addressed yet and that could help to further improve clarity of presentation (e.g. size of amino acid labelling in Figure 7A).

---

## [Author Response]

Essential revisions:1) The negative stain EM data (Figure 4F-I) needs to be improved. The blobs shown in Figure 4F are similar to those observed with badly behaved samples in EM and there is currently no evidence they correspond to the Alp14 structure. The authors would need to be able to show 2D averages of the WT sample in order to use negative stain analysis and compare the averages to projections of the crystal structure. The particles in images in H and I are too close together. There is no evidence that the particles in the white boxes correspond to one Alp14 complex. The data would need to be collected at lower dilutions so individual particles are clear. It would also be important to provide 2D classification on these images.We carried out negative electron microscopy, 2D classification, and projection-matching analyses to support the existence of the TOG square in wt-Alp14 and the nature of defects in TOG arrays in the INT1, INT2 and INT1+2 mutants in complex with tubulin, described in the revised figure 4 and its figure supplements. The TOG square organization is evident in 4:2 Alp14-ab-tubulin negative stain images. We have captured images and 2D-class averages for particles that match the conformation of TOG square assemblies with and without tubulin, which likely formed due to dissociation of ab-tubulins from Alp14 on the grids. The INT1, INT2:tubulin complexes displays a pre-polymerized (corner-bent-like) organization (as found for one subunit in a TOG square) with two non-polymerized ab-tubulins, or an unfurled post-polymerized TOG1-TOG2 bound to two ab-tubulin in a polymerized assembly, as well as isolated TOG1-tubulin complexes,.while INT1+2 shows only isolated TOG-ab-tubulin complexes. Taken together, these data show that the TOG square assembly forms in wt-Alp14, that the interfaces stabilize this conformation, and that the inactivation of these interfaces in INT1, INT2 and INT1+2 leads to specific forms of disassembled or polymerized TOG array bound ab-tubulin assemblies. The data also explain why INT1 subunits remain capable of unfurling activity.2) The evidence for the square form of the Alp14/tubulin complex playing a role in microtubule polymerization should be presented in this manuscript (particularly given that the reviewers did not support publication of the accompanying manuscript in eLife – see the Decision regarding the accompanying manuscript). The authors should explain why a rather large number of mutations were introduced to disrupt the interfaces. The reviewers expected that a smaller number of mutations could have been sufficient and wondered whether effects other than disrupting the interfaces could lead to the defects in microtubule polymerisation.

We believe that the functional studies of inactivated square assembly interface mutants, currently presented by Cook et al., do not fit within the scope or questions posed in the Nithianantham et al., manuscript. We submitted a revision of the manuscript by Cook et al., for reconsideration by the reviewers at *eLife* (see above). Those data fit within the context of the other data described by Cook et al., which focus on the dynamic cycle of processive microtubule plus-end tracking and the microtubule polymerase cycle. The addition of three figures (and associated figure supplements) would distract from the structural studies that are the focus of the present investigation, and lengthen the current manuscript, which is already fairly long.

3) In parts the amount of the presented data are rather overwhelming particularly given that not all of the data are described. The presentation of these data needs to be improved. The data shown in Figure 1 and the related supplementary figures are only partly described. Therefore, this part of the manuscript requires hours of work to really understand which data support which claims. A better way to present these data could be to present in the main Figure 1: a graph summarising the most important MALS data for some complexes without Darpin (which ultimately are the strongest evidence for binding stoichiometries), Figure 1E and G; and present everything else in supplementary figures including a clear description/summary in their legends of what one sees in these figures.

In the current manuscript, we revised the Results section to better describe the biochemical analyses of affinity and stoichiometry of tubulin binding to a variety of TOG array constructs. We have also streamlined the presentation of the experiments and conclusions in the Results section. We also revised the figure legends and supplementary figure legends to maximize clarity.

4) The presentation of the model in the Discussion section needs improvement. The current description of the model in Figure 8 gives the impression that the model is very speculative, probably because the description of the model is combined with the implications. The authors are encouraged to first describe the model in its entirety, then reiterate the supporting evidence for it, and only then discuss the implications.

We have fully rewritten the Discussion section to present the steps of the microtubule polymerase model and the implications of the mechanism. We have also added a section describing the questions that arise from this model, motivating the studies described by Cook et al. These questions help clarify the importance of the dynamic studies of microtubule polymerases that are presented by Cook et al.

5) A recent publication (Geyer et al., 2018) provides data and a model that do not appear to be in agreement with the model proposed here. The authors should discuss these recently published findings and adjust claims as appropriate in the light of these new data.

We compared our studies with those of Geyer et al., 2018 throughout both manuscripts. All-TOG1 and allTOG2 stu2 constructs showed a level of activity intermediate between those of single TOG1- or TOG2inactivated mutants and wt-Stu2 protein. Some observations made by Geyer et al. about the defects in TOG1- and TOG2-inactivated mutants and the role of dimerization are fairly consistent with data reported by Cook et al., (see our revised Cook et al. manuscript), while many other observations are not compatible. In Cook et al., we detected a dwell time (different from that reported by Geyer et al.,) for wt-Alp14 at MT plus-ends using stable fluorescent dyes. Generally, our two manuscripts focus on the higher-order organization of TOG arrays, while Geyer et al., focused on the interaction between a Ser/Lys positively charged region and the TOG arrays. Our data point to substantially different conclusions, some of which are not compatible with the conclusions of Geyer et al. We describe these issues and compare Geyer et al., in the revised Nithianantham et al., and Cook et al., manuscripts.

[Editors' note: further revisions were requested prior to acceptance, as described below.]

The manuscript has been improved. Particularly the improved negative stain EM part has strengthened the manuscript and has satisfactorily addressed a major concern of the reviewers. All reviewers also welcome the other changes, particularly the editorial changes that have improved clarity of presentation in most parts. The reviewers accept the decision of the authors not to combine the biochemical data demonstrating functional importance of the interfaces involved in 'square assembly' formation, with the structural data presented in this manuscript, although it would have strengthened the arguments made.But there are some remaining issues that need to be addressed before acceptance.Despite parts of the Discussion section being clearer now, all reviewers agree that the Discussion section can be considerably more concise. The discussion of the Cook et al. paper can be deleted, and the discussion of the Geyer et al., paper should be more balanced. The statement "all-TOG1 or TOG2 chimeric constructs retained an MT polymerase activity level that was substantially lower than native Stu2" seems to be an oversimplification, since Geyer et al. reported a 84% activity level for the TOG2-TOG2 construct, which is remarkably close to the wildtype level.We therefore invite you submit a revised version of your manuscript with a more balanced and importantly substantially shorter discussion, focusing on the main points relevant to this study. We also ask to consider addressing previously stated minor concerns that have not been addressed yet and that could help to further improve clarity of presentation (e.g. size of amino acid labelling in Figure 7A).

We appreciate the positive comments regarding the revised version. In this further revised manuscript version we have made the requested changes to Figure 7 and the following editorial changes regarding the Discussion section:

1) We present a significantly shortened and streamlined Discussion section.

2) We have removed sections related to Cook et al.

3) We have added the Title heading “Other MT polymerase models” where we added a revised more balanced our discussion of Geyer et al., 2018, addressing suggestions made by the reviewers.

4) We have removed the Conclusion section.